# AmortizedPeriod: Attention-based Amortized Inference for Periodicity Identification

**Hang Yu, Cong Liao, Ruolan Liu, Jianguo Li,*Yun Hu, Xinzhe Wang**

Ant Group, Hangzhou, 310013, China

## ABSTRACT

Periodic patterns are a fundamental characteristic of time series in natural world, with significant implications for a range of disciplines, from economics to cloud systems. However, the current literature on periodicity detection faces two key challenges: limited robustness in real-world scenarios and a lack of memory to leverage previously observed time series to accelerate and improve inference on new data. To overcome these obstacles, this paper presents AmortizedPeriod, an innovative approach to periodicity identification based on amortized variational inference that integrates Bayesian statistics and deep learning. Through the Bayesian generative process, our method flexibly captures the dependencies of the periods, trends, noise, and outliers in time series, while also considering missing data and irregular periods in a robust manner. In addition, it utilizes the evidence lower bound of the log-likelihood of the observed time series as the loss function to train a deep attention inference network, facilitating knowledge transfer from the seen time series (and their labels) to unseen ones. Experimental results show that AmortizedPeriod surpasses the state-of-the-art methods by a large margin of $28.5\%$ on average in terms of micro $F_1$-score, with at least $55\%$ less inference time.[1]

## 1 INTRODUCTION

Periodic patterns are omnipresent in this physical world, since the cyclic movements of the earth and the moon give rise to recurring events, and further influence the behavior of life on this planet. Identifying periods in real-world time series and investigating their causes shed light upon understanding how the world works, and further aid in more informed and accurate modeling and decision-making. Indeed, periodicity detection plays a pivotal role in time series-related tasks, such as classification (Orsenigo & Vercellis, 2010), clustering (Aghabozorgi et al., 2015), decomposition (Wen et al., 2019; 2020), anomaly detection (Zhao et al., 2019), and forecasting (Oreshkin et al., 2019; Liu et al., 2022; Ni et al., 2023). It thereby finds applications and permeates the literature in various domains, such as economy and finance (Nerlove et al., 2014), physics and earth science (Kong et al., 2020), biology and neuroscience (Jing et al., 2022), database and cloud operations (Rzadca et al., 2020), etc.

The large body of literature on periodicity detection can be broadly divided into four groups. The first group identifies periods in the time domain via autocorrelation or partial autocorrelation (Elfeky et al., 2004; Breitenbach et al., 2023), whereas the second exploits discrete Fourier transform (DFT) and finds periods in the frequency domain (Vlachos et al., 2004; Bauer et al., 2020). The third one further combines the time and frequency methods and allows them to complement each other (Vlachos et al., 2005; Wen et al., 2021). However, both autocorrelation and DFT are not originally intended for periodicity detection, and so their use requires arduous preprocessing and postprocessing steps to alleviate their deficiencies. Instead, the fourth group (Tenneti & Vaidyanathan, 2015; 2016) projects the time series onto the orthogonal Ramanujan subspaces (Vaidyanathan, 2014a;b), each representing an exact period, and detects periods by identifying projections with significant energy. Despite the progress, the practical utility of the above approaches is still hindered by two limitations:

**Limited robustness**: In real time series, multiple periodic components are often interlaced with each other, and are further contaminated with trend, noise, outliers, and incomplete observations. Moreover, there may exist irregular periods, such as the monthly pattern. Note that the number of

---

*Corresponding author.

[1]Code is available at https://github.com/alipay/AmortizedPeriod.

days often differs in consecutive months and monthly periods are crucial for real applications such as resource scheduling. It is thus imperative to design a periodicity detection approach that is robust to 1) multiple and 2) possibly irregular periods as well as 3) trend, 4) noise, 5) outliers, and 6) missing data. Unfortunately, the existing approaches only address some of these requirements (cf. Appendix B). In particular, none of them explicitly consider irregular periods. Additionally, they typically tackle the desiderata sequentially in a pipeline, and may suffer from the issue of error accumulation.

**Memorylessness**: In the aforementioned approaches, each time series is processed in isolation and the inferences are never reused. More specifically, even if the same time series has been analyzed before, analyzing it in the future still requires the same amount of computational cost. Consequently, these methods may incur a daunting computational burden when tackling the huge number of time series that can be found in real applications nowadays (Rzadca et al., 2020; Wu et al., 2023). This problem is exacerbated for some methods in the third and the fourth group, as they are more computationally intensive than those in the first two groups. Furthermore, memorylessness indicates that labeling information from the past cannot help to improve the performance of these algorithms in the future, even though human feedback can be available every now and then.

To lift these limitations, we propose AmortizedPeriod, an amortized variational inference based method for periodicity detection. Amortized inference can be viewed as a marriage of Bayesian statistics and deep learning (Zhang et al., 2018; Ganguly et al., 2022). The Bayesian paradigm offers a flexible way to jointly model the period, trend, noise, and outlier components and accounts for the uncertainties resulting from these components as well as the missing data. The deep learning-based inference network further maps the observed time series to the parameters of the variational periodicity distributions. By maximizing the evidence lower bound (ELBO) of the log-likelihood of the observed time series in a self-supervised manner, the resulting inference model learns to mimic the effect of probabilistic inference, such that on testing data, we can enjoy the benefits of probabilistic modeling without paying a further cost for inference from scratch. Moreover, the labeling information can be incorporated to train the inference model in a semi-supervised fashion. Viewed another way, the periodicity detection problem can be recast as a multi-class classification problem. The inference model is the classifier and is trained using the proposed self/semi-supervised framework. In a nutshell, the Bayesian aspect of AmortizedPeriod addresses the challenge of limited robustness, while the deep learning part tackles the issue of memorylessness. Our contribution is threefold:

- We propose a novel method for periodicity detection named AmortizedPeriod. To the best of our knowledge, AmortizedPeriod is the first model that can perform period learning in the presence of multiple and irregular periods as well as trends, noise, outliers, and missing data, whereas existing methods only address some of these problems (see Appendix B).
- We propose an inference network based on cross-channel attention and derive the ELBO as the loss to train this network in a self/semi-supervised manner. As a result, AmortizedPeriod is also a fresh attempt to leverage the knowledge from the previously observed time series (and their labels) to accelerate and improve the inference on new data.
- We conduct extensive experiments on four datasets, and find that AmortizedPeriod outperforms the state-of-the-art methods by $28.5\%$ on average with above $55\%$ less amount of inference time. Moreover, a relatively small fraction of labels (e.g., $15\%$) can significantly boost its performance.

## 2 RELATED WORKS

As mentioned in the introduction, existing methods for periodicity detection can be categorized into four groups. The first group resorts to the (partial) autocorrelation function (ACF) in the time domain that finds the local maxima of the similarity between the shifted and unshifted version of a time series (Elfeky et al., 2004; Wang et al., 2005; 2006; Toller & Kern, 2017; Breitenbach et al., 2023). However, this group tends to overlook smaller periods while emphasizing larger ones (Vlachos et al., 2005). They also struggle with multiple periods. In this case, the least common multiple (LCM) of the hidden periods represents the actual period, which can be significantly greater than the data length and thus may not appear in the local maxima of the ACF at all. Additionally, the effectiveness of these methods can be hampered by noise and outliers in the time series.

The second group of methods instead utilizes the DFT and detects periods based on the periodogram in the frequency domain (Vlachos et al., 2004; Tominaga, 2010; Li et al., 2010; Drutsa et al., 2017; Bauer et al., 2020). Nevertheless, the accuracy of these methods diminishes for larger periods, in contrast to the ACF (Vlachos et al., 2005). Furthermore, the periodogram aims to provide the Fourier terms that best reconstruct a time series when added together, and therefore may produce numerous

false periods when trying to approximate the shape of a non-sinusoidal time series. Finally, similar to the first group, these methods are sensitive to trend changes and outliers.

The third group of methods intends to borrow the strength of both time and frequency methods (Vlachos et al., 2005; Toller et al., 2019; Puech et al., 2020; Wen et al., 2021; 2023). They typically first select a list of candidates in the frequency domain using the DFT, and then identify the exact period in the time domain using the ACF. The fundamental idea is that a valid period revealed by the periodogram should correspond to a peak in the ACF. RobustPeriod (Wen et al., 2021) further enhances the robustness by first taking a preprocessing step that removes the trend, noise, and outliers using the Hodrick-Prescott filter, then decomposing the time series into multiple scales via discrete wavelet transform to balance the time and frequency resolution, and finally detecting and validating the period in each scale respectively based on Huber-periodogram and Huber-ACF where the Huber loss promotes robustness. Recent advancements in this field also address the presence of (block) missing data (Wen et al., 2023). While the third group outperforms the previous ones, each stage (e.g., denoising, decomposition, detection, and validation) is processed sequentially in a pipeline, implying that biases in one stage can negatively impact the subsequent stages. Moreover, the performance of each stage relies on manually chosen hyperparameters, posing a practical challenge in their selection.

The aforementioned difficulties largely stem from the inadequacy of both the ACF and the DFT for detecting periods. As a remedy, the fourth category of approaches (Tenneti & Vaidyanathan, 2015; 2016; Deng & Han, 2017; Zhang et al., 2020) utilizes the Ramanujan transform, which is specifically tailored for periodicity detection. This technique involves projecting the time series onto the orthogonal Ramanujan subspaces, each representing a distinct period. Identifying periods is then posed as a linear regression problem, whose objective is to select the Ramanujan subspaces that can best explain the time series. Despite these advancements, these techniques still exhibit limited robustness and are not able to transfer knowledge from previously observed time series to new ones.

## 3 AMORTIZEDPERIOD

The limited robustness observed in previous studies can be attributed to the neglect of non-periodic components, such as trends, noise, and outliers. While certain methods, like RobustPeriod, acknowledge these components, they eliminate them individually in a deterministic and sequential way, leading to error accumulation. Moreover, the performance of each stage is sensitive to the choice of the corresponding hyperparameters. To overcome these difficulties, we propose a compact Bayesian generative model that offers flexibility in modeling the interaction between all components as well as their corresponding hyperparameters in a joint and probabilistic manner. On the other hand, existing approaches derive period estimates for each time series from scratch by solving optimization problems iteratively. Moreover, even if we correct the estimated periods for a time series, applying existing methods would still yield the same erroneous results. To provide a remedy, we develop an attention-based inference network that directly maps observed time series to the posterior distribution of different components and the hyperparameters, including the distribution of the periods. As a result, the inference model can provide period distributions for new time series without iterative optimization. Moreover, the inference model can be trained using labeling information, as it can memorize the labels. In the sequel, before delving into the generative and inference model, we first introduce the Ramanujan subspaces, which form the foundation of AmortizedPeriod.

### 3.1 RAMANUJAN SUBSPACES AND PERIODICITY IDENTIFICATION

The definition of Ramanujan subspaces is cumbersome to describe and not illuminating, so we present instead the definition of $q$-periodic time series as a starting point. From there, we proceed to discuss the properties of Ramanujan subspaces relevant to periodicity detection and define exactly $q$-periodic time series. We refer the readers to Appendix C for the formal definition of Ramanujan subspaces.

**Definition 1.** *(q-periodic) A time series $s(t)$ with time stamp $t$ of length $L$ ($1 \leq t \leq L$) is q-periodic if q is the smallest integer such that $s(t + q) = s(t)$ for all t, and q is called the period of $s(t)$.*

This definition eliminates the ambiguity that a time series of period $q$ is also of a period that is a multiple of $q$, which plagues the time-domain methods for periodicity detection. However, another problem comes along with this definition, that is, a time series with a period of 6, for instance, may be further decomposed into two periodic components with periods 2 and 3 respectively, making it necessary to detect these fundamental exact periods in practical scenarios. To attack this problem, Vaidyanathan (2014a;b) proposed the Ramanujan subspaces, which construct a distinct linear subspace

$\mathcal{S}_q$ for each period $q$. The time series spanned by a subspace $\mathcal{S}_q$ can only possess a period of $q$ and cannot be decomposed into smaller periods. Moreover, two arbitrary subspaces $\mathcal{S}_{q_1}$ and $\mathcal{S}_{q_2}$ are orthogonal to each other if $q_1 \neq q_2$. As a result, we can define the exact period as:

**Definition 2.** *(exactly q-periodic (Muresan & Parks, 2003)) A time series $s(t)$ is of exact period q if $s(t)$ is in the Ramanujan subspace $\mathcal{S}_q$ and the projection of $s(t)$ onto $\mathcal{S}_{\bar{q}}$ is zero for all $\bar{q} < q$.*

Let $\boldsymbol{R}_q$ be the linear basis for $\mathcal{S}_q$ and $\boldsymbol{D} = [\boldsymbol{R}_1, \cdots, \boldsymbol{R}_{P_{\max}}]$ the resulting Ramanujan dictionary with a maximum period of $P_{\max}$. We can ascertain the periods of a time series $s(t)$ by decomposing it as $\boldsymbol{s} = \boldsymbol{D}\boldsymbol{\alpha}$, where $\boldsymbol{\alpha}$ is the coefficient vector. Ideally, $\boldsymbol{\alpha}$ should have non-zero entries only at locations where the periodic components of $\boldsymbol{s}$ reside. Estimating $\boldsymbol{\alpha}$ given $\boldsymbol{s}$ and $\boldsymbol{D}$ can be formulated as a sparse vector recovery problem, such as basis pursuit or lasso (Tenneti & Vaidyanathan, 2016).

### 3.2 Generative Model

Unfortunately, the above formulation is susceptible to the presence of trends, noise, outliers, and incomplete observations that often coexist with periodic components in time series, as well as the hyperparameter promoting sparsity in basis pursuit or lasso. Conversely, Bayesian models or networks offer a compact, flexible, and interpretable integration of various components and their associated hyperparameters in time series, while also accounting for the accompanying uncertainty. However, their application in the context of periodicity identification remains unexplored. To bridge the gap, we propose a reformulation of the periodicity detection problem from the Bayesian perspective.

Suppose that a time series $\boldsymbol{x} = \boldsymbol{x}(1 : L)$ after normalization can be decomposed into periods $\boldsymbol{s}$, trend $\boldsymbol{\tau}$, noise $\boldsymbol{\epsilon}$, and outliers $\boldsymbol{\delta}$, that is, $\boldsymbol{x} = \boldsymbol{s} + \boldsymbol{\tau} + \boldsymbol{\epsilon} + \boldsymbol{\delta}$. Next, we introduce the prior distribution for each component individually. More details on the priors can be found in Appendix D.

**The periodic component $\boldsymbol{s}$**: It follows from Section 3.1 that $\boldsymbol{s} = \boldsymbol{D}\boldsymbol{\alpha}$, where $\boldsymbol{D}$ is the Ramanujan dictionary and $\boldsymbol{\alpha}$ represents the coefficient vector. $\alpha_j \neq 0$ indicates that the period characterized by $\boldsymbol{D}_j$ exits in the time series $\boldsymbol{x}$. To distinguish between the zero and non-zero entries in $\boldsymbol{\alpha}$, we resort to the horse-shoe prior (Carvalho et al., 2009), which can be interpreted as a scale mixture of Gaussians:

$$p(\alpha_j | \sigma_j, \nu) = \mathcal{N}(\alpha_j; 0, \nu^2 \sigma_j^2), \qquad p(\sigma_j) = C^+(0, 1), \qquad (1)$$

where $C^+(0, 1)$ is a standard half-Cauchy distribution on the positive reals $\mathbb{R}^+$, and $\nu$ and $\sigma_j$ respectively denote the global and local shrinkage parameters. The global shrinkage parameter $\nu$ shrinks all $\alpha_j$ to zero, while the heavy-tailed half-Cauchy priors for the local shrinkage parameters $\sigma_j$ allow some $\alpha_j$ to escape from the shrinkage. The resulting $\boldsymbol{\alpha} = [\alpha_1, \cdots, \alpha_\Phi]$ would therefore be sparse. One attractive property of the horse-shoe distribution is that the shrinkage weight, $1/(1 + \sigma_j^2)$, has a U-shaped density (i.e., horse-shoe shape) $\mathrm{Beta}(0.5, 0.5)$, which is a Beta distribution with shape parameters $a = b = 0.5$, as shown in Appendix D.2. It reaches the lowest value at 0.5 but is unbounded at 0 and 1, indicating that this prior prefers $\sigma_j^2$ to be either very small or very large and can well separate the zero and nonzero values in $\boldsymbol{\alpha}$. To facilitate the amortized inference, we innovatively reparameterize the horse-shoe prior as:

$$p(\alpha_j | \omega_j, \lambda) = \mathcal{N}\left(\alpha_j; 0, \left(\lambda \frac{1 - \omega_j}{\omega_j}\right)^{-1}\right), \qquad p(\omega_j) = \mathrm{Beta}(\omega_j; 0.5, 0.5) \qquad (2)$$

where $\lambda = 1/\nu^2$ and $\omega_j = \sigma_j^2/(1 + \sigma_j^2)$. As a result, the expectation of $\omega_j$ provides an indicator of whether $\alpha_j = 0$. We then specify the hyperprior on $\lambda$ to be a Gamma distribution $\mathrm{Gamma}(a_\lambda, b_\lambda)$, where the shape $a_\lambda$ and rate $b_\lambda$ are set small (e.g., $10^{-4}$) such that the hyperprior is non-informative.

**The trend component $\boldsymbol{\tau}$**: The trend is assumed to vary smoothly across time. Therefore, we employ the thin-plate model (a.k.a the second-order intrinsic Gauss-Markov random field) (Rue & Held, 2005; Yu & Dauwels, 2016) to describe this behavior, since it penalizes the second-order difference (i.e., curvature). Concretely, we model the second-order differences as a Gaussian distribution: $\Delta^2 \boldsymbol{\tau}(t) \sim \mathcal{N}(0, \beta_1^{-1})$. In other words, the density function of a thin-plate model is (cf. Appendix D.3):

$$p(\boldsymbol{\tau}) \propto \exp\left(-\frac{\beta_1}{2} \sum_{t=2}^{L-1} \left(\boldsymbol{\tau}(t-1) - 2\boldsymbol{\tau}(t) + \boldsymbol{\tau}(t+1)\right)^2\right) \propto \exp\left(-\frac{\beta_1}{2} \boldsymbol{\tau}^T \boldsymbol{K}_{tp} \boldsymbol{\tau}\right), \qquad (3)$$

where the smoothness parameter $\beta_1$ controls the curvature of the trend over time, and $\boldsymbol{K}_{tp}$ is the weighted Laplacian matrix. It can be observed from (3) that the thin-plate model is invariant to the addition of a constant, and more importantly, a linear function of time $t$. As such, this prior can accommodate the linear trends without penalty. Moreover, as the Ramanujan dictionary $\boldsymbol{D}$ has a constant subspace $\mathcal{S}_0$ that can describe the overall mean level of the time series, we specify the mean

of the trend component $\boldsymbol{\tau}$ to be 0. The resulting prior on $\boldsymbol{\tau}$ can be expressed as:

$$p(\boldsymbol{\tau}|\beta_0, \beta_1) \propto \exp\left(-\frac{1}{2}\boldsymbol{\tau}^T(\beta_0\boldsymbol{I} + \beta_1\boldsymbol{K}_{tp})\boldsymbol{\tau}\right), \qquad (4)$$

where $\beta_0$ measures how close to zero $\boldsymbol{\tau}(t)$ is for all $t$. We again impose non-informative Gamma hyperprior $\mathrm{Gamma}(10^{-4}, 10^{-4})$ on $\beta_0$ and $\beta_1$.

**The outliers $\boldsymbol{\delta}$ and the noise $\boldsymbol{\epsilon}$:** Typically, most of the elements in the outliers $\boldsymbol{\delta}$ are 0, suggesting that $\boldsymbol{\delta}$ is sparse. To this end, we impose the Student's $t$-distribution on $\boldsymbol{\delta}(t)$, which is commonly used as a shrinkage prior in the literature of sparse Bayesian learning (Tipping, 2001; Yu et al., 2020a). Equivalently, it can be written in a hierarchical manner as (cf. Appendix D.1):

$$p(\boldsymbol{\delta}(t)|v_{\delta(t)}) = \mathcal{N}(\boldsymbol{\delta}(t); 0, v_{\delta(t)}), \qquad p(v_{\delta(t)}) = \mathrm{Inv\text{-}Gamma}(v_{\delta(t)}; a_{\delta(t)}, b_{\delta(t)}), \qquad (5)$$

where $\mathrm{Inv\text{-}Gamma}(v_{\delta(t)}; a_{\delta(t)}, b_{\delta(t)})$ denotes the inverse Gamma distribution with shape parameter $a_{\delta(t)}$ and scale parameter $b_{\delta(t)}$. Again, we set $a_{\delta(t)} = b_{\delta(t)} = 10^{-4}$ to make the inverse gamma hyperprior non-informative. On the other hand, we assume that $\boldsymbol{\epsilon} \sim \mathcal{N}(0, v_\epsilon \boldsymbol{I})$ is the Gaussian white noise and further posit the non-informative conjugate inverse Gamma prior on $v_\epsilon$. Note that we use the $t$-prior instead of the horse-shoe prior for the period coefficient vector $\boldsymbol{\alpha}$ here. The latter is more complex but comes along with a Beta distributed variable $\omega_j$ that indicates whether the corresponding period exists, which is required for the sake of periodicity detection. However, since our main focus is not on whether the outlier exists (i.e., $\delta(t) = 0$ or not), the simpler $t$-prior is a more suitable choice.

Since the outliers and the noise are independent, the distribution of their sum $\boldsymbol{\epsilon}(t) + \boldsymbol{\delta}(t)$ can be simplified as $\mathcal{N}(0, v_\epsilon + v_{\delta(t)})$. As a consequence, the distribution of the observed time series $\boldsymbol{x}$ given the latent components can be expressed as:

$$p(\boldsymbol{x}|\boldsymbol{\alpha}, \boldsymbol{\tau}, v_\epsilon, \boldsymbol{v_\delta}) = \mathcal{N}(\boldsymbol{x}; \boldsymbol{D}\boldsymbol{\alpha} + \boldsymbol{\tau}, v_\epsilon \boldsymbol{I} + \mathrm{diag}(\boldsymbol{v_\delta})), \qquad (6)$$

where $\boldsymbol{I}$ denotes the identity matrix, and $\mathrm{diag}(\boldsymbol{v_\delta})$ is a diagonal matrix with $v_{\delta(t)}$ on the diagonal.

**The overall generative model:** Taken together, the overall Bayesian network w.r.t. (with regard to) all random variables can be factorized as:

$$\begin{aligned}
p(\boldsymbol{x}, \boldsymbol{\alpha}, \boldsymbol{\omega}, \lambda, \boldsymbol{\tau}, \beta_0, \beta_1, v_\epsilon, \boldsymbol{v_\delta}) = {} & p(\boldsymbol{x}|\boldsymbol{\alpha}, \boldsymbol{\tau}, v_\epsilon, \boldsymbol{v_\delta})p(\boldsymbol{\alpha}|\boldsymbol{\omega}, \lambda)p(\boldsymbol{\omega})p(\lambda) \\
& \cdot p(\boldsymbol{\tau}|\beta_0, \beta_1)p(\beta_0)p(\beta_1)p(v_\epsilon)p(\boldsymbol{v_\delta}). \qquad (7)
\end{aligned}$$

We further consider a more general scenario where multiple time series $\boldsymbol{x}^{\{i\}}$ for $i = 1, \cdots, N$ share the same periods (as indicated by $\boldsymbol{\omega}$) but possibly with different values of the coefficients $\boldsymbol{\alpha}^{\{i\}}$. In other words, the zero pattern of $\boldsymbol{\alpha}^{\{i\}}$ is the same for all $i$, but the non-zero values in $\boldsymbol{\alpha}^{\{i\}}$ can differ. The remaining three components $\boldsymbol{\tau}^{\{i\}}$, $\boldsymbol{\epsilon}^{\{i\}}$, and $\boldsymbol{\delta}^{\{i\}}$ can also be different. The resulting Bayesian network can be factorized as:

$$\begin{aligned}
p(\boldsymbol{x}^{\{1:N\}}, \boldsymbol{\alpha}^{\{1:N\}}, \boldsymbol{\omega}, \lambda, \boldsymbol{\tau}^{\{1:N\}}, \beta_0, \beta_1, v_\epsilon^{\{1:N\}}, \boldsymbol{v_\delta}^{\{1:N\}}) = {} & \prod_{i=1}^{N}\left[p(\boldsymbol{x}^{\{i\}}|\boldsymbol{\alpha}^{\{i\}}, \boldsymbol{\tau}^{\{i\}}, v_\epsilon^{\{i\}}, \boldsymbol{v_\delta}^{\{i\}})\right. \\
\cdot p(\boldsymbol{\alpha}^{\{i\}}|\boldsymbol{\omega}, \lambda)p(\boldsymbol{\tau}^{\{i\}}|\beta_0, \beta_1)p(v_\epsilon^{\{i\}})p(\boldsymbol{v_\delta}^{\{i\}})\Big] & p(\boldsymbol{\omega})p(\lambda)p(\beta_0)p(\beta_1), \qquad (8)
\end{aligned}$$

where the priors on $\boldsymbol{\alpha}^{\{i\}}$ is the same for all $i$ and guarantees that the estimated periods are the same for $\boldsymbol{x}^{\{1:N\}}$. The corresponding factor graph representation (Kschischang et al., 2001) of (8) is depicted in Fig. 1(a). Note that the above equation amounts to (7) when $N = 1$. This formulation is widely applicable to periodicity detection of time series in various fields in practice (Zhang et al., 2020), such as the electricity usage and the workloads of data centers, to name a few. Moreover, real-world time series are often associated with a given fundamental period (e.g., daily period), and other periods (e.g., weekly and monthly periods) are a multiple of this fundamental period. Under this scenario, the $i$-th time point in consecutive days can form a new time series $\boldsymbol{x}^{\{i\}}$. Suppose that there are $N$ time points in a day, and we can obtain $N$ time series $\boldsymbol{x}^{\{1:N\}}$ with common periods.

## 3.3 Inference Model

**The ELBO:** Owing to the U-shaped prior on $\omega_j$, the posterior distribution of $\omega_j$ also follows a U-shape, as observed in (Yu et al., 2019; Chen et al., 2023). As a result, our ultimate goal is to detect the periodicity by checking how close to 1 $\omega_i$ is given the observed time series $\boldsymbol{x}^{\{1:N\}}$, namely, $p(\omega_j = 1|\boldsymbol{x}^{\{1:N\}})$. To this end, we need to compute the exact posterior $p(\boldsymbol{\alpha}^{\{1:N\}}, \boldsymbol{\omega}, \lambda, \boldsymbol{\tau}^{\{1:N\}}, \beta_0, \beta_1, v_\epsilon^{\{1:N\}}, \boldsymbol{v_\delta}^{\{1:N\}}|\boldsymbol{x}^{\{1:N\}})$. Since this posterior is intractable, we instead approximate it with an inference model $q(\boldsymbol{\alpha}^{\{1:N\}}, \boldsymbol{\omega}, \lambda, \boldsymbol{\tau}^{\{1:N\}}, \beta_0, \beta_1, v_\epsilon^{\{1:N\}}, \boldsymbol{v_\delta}^{\{1:N\}})$, which

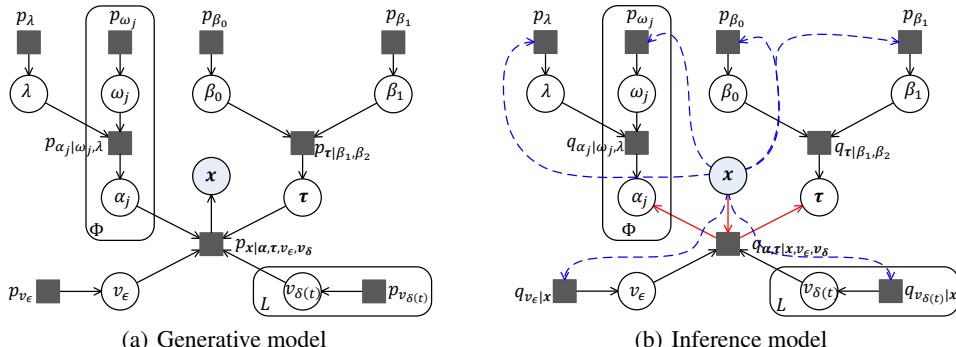

|   | (a) Generative model | (b) Inference model |
| --- | --- | --- |

Figure 1: Factor graph representation (Kschischang et al., 2001) of AmortizedPeriod. The circle and square nodes respectively denote variables and factors in the generative (8) and inference (10) model. The rectangular plate labeled $\Phi$ or $L$ means that there are $\Phi$ or $L$ subgraphs of this kind. We abuse the edge direction to denote the sampling process in the two models. Note that the inference model maintains the dependence structure of the generative model to better approximate the true posterior. It further includes the addition (blue dashed lines) and alteration (red solid lines) of several edges to ensure that all latent variables are dependent on the observation $\boldsymbol{x}$.

is a probabilistic encoder outputting the latent variables given the observations, using the framework of amortized variational inference (Zhang et al., 2018; Ganguly et al., 2022), in analogy to the VAE (Kingma & Welling, 2014). The parameters of both the generative and the inference model can be jointly estimated by maximizing the evidence lower bound (ELBO) $\mathcal{L}$ of $\log p(\boldsymbol{x}^{\{1:N\}})$:

$$\mathcal{L} = \mathbb{E}_q\Big[\log p(\boldsymbol{x}^{\{1:N\}}, \boldsymbol{\alpha}^{\{1:N\}}, \boldsymbol{\omega}, \lambda, \boldsymbol{\tau}^{\{1:N\}}, \beta_0, \beta_1, v_\epsilon^{\{1:N\}}, \boldsymbol{v}_{\boldsymbol{\delta}}^{\{1:N\}})\Big] + \mathbb{H}_q, \qquad (9)$$

where $\mathbb{E}_q$ denotes expectation over the $q$ distribution, and $\mathbb{H}_q$ denotes the entropy of the $q$ distribution.

**The $q$ distributions**: However, to guarantee that the inference process is tractable, the VAE (Kingma & Welling, 2014) and the commonly-used mean-field amortized inference method (Zhang et al., 2018; Ganguly et al., 2022) typically assumes that all latent variables are independent. Consequently, the approximation accuracy is limited, especially when the latent variables in the original generative model are highly dependent such as in our model (8) with a hierarchical structure, as pointed out in (Turner & Sahani, 2011; Ranganath et al., 2016). As a remedy, we design a structured inference network (Lin et al., 2018; Ambrogioni et al., 2021; Agrawal & Domke, 2021; Rouillard & Wassermann, 2022) that can incorporate the dependence structure of the generative model into the inference model in a straightforward manner. As shown in the factor graph representation in Fig. 1(b), we consider an inference model that factorizes in a similar fashion to the generative model, but is conditioned on the observation $\boldsymbol{x}^{\{1:N\}}$:

$$q(\boldsymbol{\alpha}^{\{1:N\}}, \boldsymbol{\omega}, \lambda, \boldsymbol{\tau}^{\{1:N\}}, \beta_0, \beta_1, v_\epsilon^{\{1:N\}}, \boldsymbol{v}_{\boldsymbol{\delta}}^{\{1:N\}}) = \prod_{i=1}^{N}\Big[q(\boldsymbol{\alpha}^{\{i\}}, \boldsymbol{\tau}^{\{i\}}|\boldsymbol{x}^{\{i\}}, \boldsymbol{\omega}, \lambda, \beta_0, \beta_1, v_\epsilon^{\{i\}}, \boldsymbol{v}_{\boldsymbol{\delta}}^{\{i\}})$$

$$\cdot q(v_\epsilon^{\{i\}}|\boldsymbol{x}^{\{i\}})q(\boldsymbol{v}_{\boldsymbol{\delta}}^{\{i\}}|\boldsymbol{x}^{\{i\}})\Big]\prod_{j=1}^{\Phi}q(\omega_j|\boldsymbol{x}^{\{1:N\}})q(\lambda|\boldsymbol{x}^{\{1:N\}})q(\beta_0|\boldsymbol{x}^{\{1:N\}})q(\beta_1|\boldsymbol{x}^{\{1:N\}}). \qquad (10)$$

Next, we elaborate on the specification of the factors in the inference model (10). One criterion for choosing the $q$ distributions is to facilitate the computation of the expectation w.r.t $q$ in (9). The expectation is often computed via a Monte Carlo estimator based on the "reparameterization trick" (Kingma & Welling, 2014): a variate $U$ with a simple distribution that is independent of all parameters $\boldsymbol{\theta}$ of $q$ is first defined, and a reparameterization function $F$ is derived such that $F(U, \boldsymbol{\theta})$ has distribution $q$. On the other hand, the $q$ distributions should be similar to (if not the same as) the corresponding $p$ distributions for the sake of better approximation. To this end, we specify $q(\lambda|\boldsymbol{x}^{\{1:N\}})$, $q(\beta_0|\boldsymbol{x}^{\{1:N\}})$, $q(\beta_1|\boldsymbol{x}^{\{1:N\}})$, $q(v_\epsilon^{\{i\}}|\boldsymbol{x}^{\{i\}})$, and $q(\boldsymbol{v}_{\boldsymbol{\delta}}^{\{i\}}|\boldsymbol{x}^{\{i\}})$ to be log-normal distributions in order to approximate the Gamma and Inverse Gamma distributions, and $q(\omega_j|\boldsymbol{x}^{\{1:N\}})$ to be the Kumaraswamy distribution (Nalisnick & Smyth, 2017) to approximate the Beta distribution. In addition, to consider the dependence structure in the generative model, we assume that $q(\boldsymbol{\alpha}^{\{i\}}, \boldsymbol{\tau}^{\{i\}}|\boldsymbol{x}^{\{i\}}, \boldsymbol{\omega}, \lambda, \beta_0, \beta_1, v_\epsilon^{\{i\}}, \boldsymbol{v}_{\boldsymbol{\delta}}^{\{i\}})$ takes the functional form of $p(\boldsymbol{\alpha}^{\{i\}}, \boldsymbol{\tau}^{\{i\}}|\boldsymbol{x}^{\{i\}}, \boldsymbol{\omega}, \lambda, \beta_0, \beta_1, v_\epsilon^{\{i\}}, \boldsymbol{v}_{\boldsymbol{\delta}}^{\{i\}})$, which is a Gaussian distribution as proven below.

**Proposition 1.** *Given* $p(\boldsymbol{x}^{\{i\}}|\boldsymbol{\alpha}^{\{i\}}, \boldsymbol{\tau}^{\{i\}}, v_\epsilon^{\{i\}}, \boldsymbol{v}_{\boldsymbol{\delta}}^{\{i\}})$, $p(\alpha_j^{\{i\}}|\omega_j, \lambda)$, *and* $p(\boldsymbol{\tau}^{\{i\}}|\beta_0, \beta_1)$ *as defined in* (8), $p(\boldsymbol{\alpha}^{\{i\}}, \boldsymbol{\tau}^{\{i\}}|\boldsymbol{x}^{\{i\}}, \boldsymbol{\omega}, \lambda, \beta_0, \beta_1, v_\epsilon^{\{i\}}, \boldsymbol{v}_{\boldsymbol{\delta}}^{\{i\}}) = \mathcal{N}(K^{-1}h, K^{-1})$ *with precision matrix (inverse*

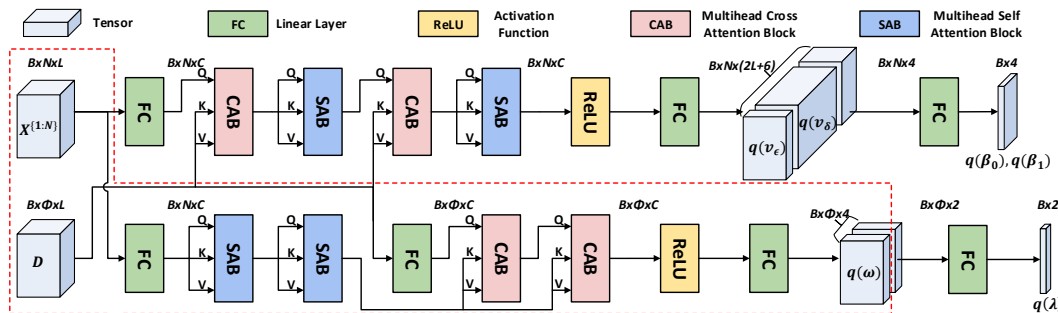

Figure 2: The inference network of AmortizedPeriod whose outputs are the parameters of the $q$ distributions. Here the $N$ time series can exchange information via the SAB as they share common periods, but the $\Phi$ basis cannot as they encode distinct periods. To detect periods, only operations inside the red dashed box are required.

*covariance) $K$ and potential vector $h$ given by:*

$$K^{\{i\}} = \begin{bmatrix} \lambda \operatorname{diag}\left(\dfrac{1-\omega_j}{\omega_j}\right) + D^T \operatorname{diag}\left(v_\epsilon^{\{i\}} + v_\delta^{\{i\}}\right)^{-1} D & D^T \operatorname{diag}\left(v_\epsilon^{\{i\}} + v_\delta^{\{i\}}\right)^{-1} \\ \operatorname{diag}\left(v_\epsilon^{\{i\}} + v_\delta^{\{i\}}\right)^{-1} D & \beta_0 I + \beta_1 K_{tp} + \operatorname{diag}\left(v_\epsilon^{\{i\}} + v_\delta^{\{i\}}\right)^{-1} \end{bmatrix},$$

$$h^{\{i\}} = \begin{bmatrix} D^T \\ I \end{bmatrix} \operatorname{diag}\left(v_\epsilon^{\{i\}} + v_\delta^{\{i\}}\right)^{-1} x^{\{i\}}.$$

*Proof.* See Appendix E. □

**The inference network**: The parameters of $q$ distributions w.r.t. $\omega$, $\lambda$, $v_\epsilon$, and $v_\delta$ are expressed as functions of the observations $x^{\{1:N\}}$ through a deep neural network, which is based on the multihead cross attention block (CAB) (Vaswani et al., 2017). To begin with, let us define the CAB:

$$\mathrm{CAB}(\mathbf{A}, \mathbf{B}) = \mathrm{LN}(\mathbf{H} + \mathrm{FFN}(\mathbf{H})), \qquad \mathbf{H} = \mathrm{LN}(\mathrm{MHA}(\mathbf{A}, \mathbf{B}, \mathbf{B}) + \mathbf{A}), \qquad (11)$$

where LN denotes layer normalization, FFN denotes the row-wise feedforward network, and $\mathrm{MHA}(\mathbf{Q}, \mathbf{K}, \mathbf{V})$ is the multihead attention layer as defined in (Vaswani et al., 2017) where $\mathbf{Q}$, $\mathbf{K}$, and $\mathbf{V}$ serve as queries, keys, and values. CAB can be interpreted as selecting keys $\mathbf{K}$ that are similar to the queries $\mathbf{Q}$ and aggregating the corresponding values $\mathbf{V}$ to update the queries. Note that $\mathrm{CAB}(\mathbf{A}, \mathbf{A})$ amounts to performing self-attention on $\mathbf{A}$, denoted as, $\mathrm{SAB}(\mathbf{A})$.

In the proposed model, the latent variables can be partitioned into two groups: those responsible for determining which bases to be used (i.e., $\omega$, and $\lambda$) and those associated with other components in the time series (i.e., $\tau$, $\beta_0$, $\beta_1$, $v_\epsilon$, and $v_\delta$). For the former group, our objective is to update the parameters for basis selection by checking the seasonality behavior of the $N$ time series. Hence, after passing the $N$ time series $x^{\{1:N\}}$ that share common periods through two layers of the self-attention block $\mathrm{SAB}(x^{\{1:N\}})$, we apply two layers of $\mathrm{CAB}(D, x^{\{1:N\}})$ in which the queries $\mathbf{Q}$ are the Ramanujan basis $D$ and the keys $\mathbf{K}$ and values $\mathbf{V}$ are the representations of the $N$ time series $x^{\{1:N\}}$ given by the last SAB. For the latter group, we aim to remove the periodic components from the time series and estimate the latent variables related to trend, noise, and outliers. To this end, we employ two $\mathrm{CAB}(x^{\{1:N\}}, D)$ layers with the $N$ time series as the queries and the basis $D$ as the keys and values. These are each followed by a self-attention block $\mathrm{SAB}(x^{\{1:N\}})$.

We remark that the attention operation is computed over different time series and bases, unlike the conventional approach of operating over the time dimension when acquiring time series representations through Transformers (Zhang & Yan, 2023). As a result, such an operation is permutation equivariant to the order of the times series and the Ramanujan bases, which is a desirable characteristic. The overall architecture of the inference network is sketched in Figure. 2. Note that during testing, we only need to compute $q(\omega_j|x^{\{1:N\}})$. Hence, only operations within the red dashed box are required.

### 3.4 SEMI-SUPERVISED LEARNING

One advantage of AmortizedPeriod is that the labels of the periods can be leveraged to train the inference network semi-supervisedly. For a time series with period of $q$, it is likely that all bases within the Ramanujan subspaces associated with $q$ and its divisors $d_q$ will be selected, as the period of a time series equals the LCM of the exact periods of all its periodic components (cf. Proposition 2). However, the specific bases chosen are uncertain and depend on the characteristics of the time series. By contrast, given the exact period and its divisors, the bases corresponding to other periods can be excluded.

Table 1: The micro $F_1$-score and run time (seconds) averaged over all time series resulting from all benchmark methods for four datasets. The first two datasets consist of synthetic data with different noise variance $v_\epsilon$ and proportion of outliers $\delta\%$. The standard deviation resulting from 5 trials is shown in Table 6 in the appendix.

| Methods | $v_\epsilon = 0.1, \delta\% = 0.05$ | | | $v_\epsilon = 1, \delta\% = 0.1$ | | | Yahoo | | | App Flow | | |
|---|---|---|---|---|---|---|---|---|---|---|---|---|
| | Train | Test | Time | Train | Test | Time | Train | Test | Time | Train | Test | Time |
| AutoPeriod | 0.29 | 0.30 | 1.76e-1 | 0.29 | 0.28 | 9.88e-2 | 0.75 | 0.73 | 1.50e-1 | 0.40 | 0.45 | 7.68e-2 |
| RobustPeriod | 0.32 | 0.30 | 5.91e2 | 0.31 | 0.29 | 4.41e2 | 0.84 | 0.82 | 1.24e2 | 0.57 | 0.56 | 3.72e2 |
| RPD | 0.24 | 0.23 | 5.39 | 0.18 | 0.17 | 5.06 | 0.83 | 0.84 | 1.17 | 0.59 | 0.60 | 8.47 |
| AmortizedPeriod | 0.80 | 0.76 | 3.23e-2 | 0.70 | 0.57 | 3.32e-2 | 0.86 | 0.87 | 3.48e-2 | 0.89 | 0.87 | 3.43e-2 |

Accordingly, let $S_q$ and $S_{d_q}$ denote the Ramanujan subspace with exact period $q$ and its divisors respectively. For each exact period $q$, we first compute the log-likelihood $\log q(\omega_j = 1|\boldsymbol{x}^{\{1:N\}})$ for all $j \in S_q \cup S_{d_q}$. We then maximize the maximum of the above set of the likelihoods, since at least one basis $D_j$ from $S_q \cup S_{d_q}$ will be chosen. On the other hand, for $j \notin S_q \cup S_{d_q}$, we maximize the minimum of the log-likelihoods $\log q(\omega_j = 0|\boldsymbol{x}^{\{1:N\}})$ for all $j$, in order to learn the inference network such that the likelihood of $\omega_j = 0$ is maximized for those non-existing periods. In summary, to conduct semi-supervised learning, we add the following two terms to the ELBO $\mathcal{L}$ (9):

$$\max_{j \in S_q \cup S_{d_q}} \left( \log q(\omega_j = 1|\boldsymbol{x}^{\{1:N\}}) \right) + \min_{j \notin S_q \cup S_{d_q}} \left( \log q(\omega_j = 0|\boldsymbol{x}^{\{1:N\}}) \right). \tag{12}$$

### 3.5 IRREGULAR PERIODS AND MISSING DATA

For irregular periods (e.g., monthly period), we assume that the true period is still regular (i.e., 31 days for the monthly period) but the observations $\boldsymbol{x}$ are irregularly sampled. We then remove those rows in the Ramanujan basis corresponding to the days that do not exist in reality, and use the remaining rows in our model. On the other hand, missing data can be handled in a natural way by ignoring the corresponding factors in the generative model. In the inference model, we also ignore the missing data when computing $q(\boldsymbol{\alpha}^{\{i\}}, \boldsymbol{\tau}^{\{i\}}|\boldsymbol{x}_O^{\{i\}}, \boldsymbol{\omega}, \lambda, \beta_0, \beta_1, v_\epsilon^{\{i\}}, \boldsymbol{v}_\delta^{\{i\}})$, where $\boldsymbol{x}_O^{\{i\}}$ denotes the observed data. This conditional distribution has a closed-form expression as explained in Appendix E. Moreover, when training the inference network in a self or semi-supervised manner, we replace missing values with zero. Consequently, the inference network can regard zero as missing values.

## 4 EXPERIMENTS

We demonstrate the usefulness of AmortizedPeriod on four datasets, including two synthetic datasets (with different noise variance $v_\epsilon$ and proportions of outliers $\delta\%$) and two real datasets (Yahoo data and App Flow data). As discussed in detail in Appendix G, the identification of periods becomes progressively less challenging for synthetic data, App Flow data, and Yahoo data. The experiment configuration can be found in Appendix H, and the complete results are presented in Appendix I-J.

We first benchmark the proposed AmortizedPeriod with three state-of-the-art (SOTA) methods, including AutoPeriod (Vlachos et al., 2005) and RobustPeriod (Wen et al., 2021) in the third group, as well as the Ramanujan Periodic Dictionary based method (Tenneti & Vaidyanathan, 2015) (referred to as RPD) in the fourth group. As mentioned in Section 2, AutoPeriod and RobustPeriod are time-frequency methods that borrow the strength of both time and frequency methods for periodicity identification. For RPD, we use the Lasso formulation in (Tenneti & Vaidyanathan, 2015) which can automatically determine whether the coefficient associated with a key in the Ramanujan dictionary is zero with soft-thresholding. We assess all methods in terms of micro $F_1$-score and the average run time. The former is commonly used to evaluate the performance of multilabel classification methods, and periodicity detection can be regarded as a multilabel classification problem. The latter measures the efficiency of the methods. In practice, the number of time series can be huge, and therefore, a fast approach is desired. The results for all four methods are listed in Table 1.

AmortizedPeriod demonstrates the highest micro $F_1$-score while maintaining the shortest run time, especially for the challenging synthetic dataset. It is worth noting that both AmortizedPeriod and RobustPeriod consider the non-periodic components in the time series (e.g., trend changes, noise, and outliers). However, AmortizedPeriod is even better than RobustPeriod, since it models all components and their associated hyperparameters jointly in a flexible probabilistic manner, whereas RobustPeriod models each component sequentially in a pipeline without fully considering their relations and requires manual determination of hyperparameters. Furthermore, the lengthy sequential pipeline in RobustPeriod significantly increases run time compared to other methods. In contrast, the inference

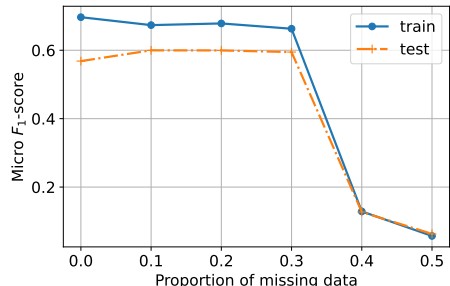 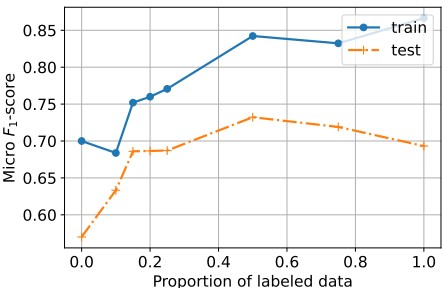

Figure 3: The micro $F_1$-score resulting from AmortizedPeriod as a function of the proportion of (a) missing data and (b) labeled data for the synthetic data with $v_\epsilon = 1$ and $\delta\% = 0.1$.

Table 2: The micro $F_1$-score for the ablation study. The standard deviation is shown in Table 7 in the appendix.

| Methods | $v_\epsilon = 0.1, \delta\% = 0.05$ | | $v_\epsilon = 1, \delta\% = 0.1$ | | Yahoo | | App Flow | |
|---|---|---|---|---|---|---|---|---|
| | Train | Test | Train | Test | Train | Test | Train | Test |
| AmortizedPeriod | 0.80 | 0.76 | 0.70 | 0.57 | 0.86 | 0.87 | 0.89 | 0.87 |
| - Trend | 0.58 | 0.55 | 0.58 | 0.48 | 0.85 | 0.86 | 0.88 | 0.86 |
| - Outliers | 0.51 | 0.48 | 0.28 | 0.29 | 0.86 | 0.86 | 0.55 | 0.50 |
| - Month | 0.71 | 0.70 | 0.53 | 0.49 | - | - | 0.85 | 0.84 |

amortization in AmortizedPeriod leads to the shortest run time, highlighting its superiority over the remaining methods that perform inference from scratch for each time series. Recall that the inference model of AmortizedPeriod is a neural network. Thus, the run time can be further shortened when the model is deployed on GPUs. On the other hand, AutoPeriod and RPD are more susceptible to the non-periodic components in the time series. Although RPD performs relatively well for Yahoo and App Flow data, where the periodic components dominate the time series (see Figure 7-8 in Appendix G), its performance declines when applied to synthetic data. By integrating the Ramanujan subspaces into the proposed framework, we significantly boost the performance of RPD.

We further investigate the impact of missing data and labeled data on AmortizedPeriod. The results are presented in Fig. 3. As demonstrated in Fig. 3(a), missing data does not significantly affect the performance of AmortizedPeriod when its proportion is less than or equal to $30\%$. However, when the proportion exceeds $30\%$, the performance deteriorates rapidly. On the other hand, leveraging labeled data to train the inference network can significantly improve the performance of AmortizedPeriod, enhancing the micro $F_1$-score by above $15\%$ for both training and testing data. Notably, for testing data, the accuracy improves faster when the first $15\%$ of labels are introduced, and stabilizes thereafter, indicating that a small proportion of labeled data can significantly enhance performance. This is especially relevant in practice, where human experts can only label a small fraction of time series data, yet such data can benefit AmortizedPeriod considerably. Other methods, however, cannot take advantage of the labeling information.

Finally, we conduct an ablation study to verify the effectiveness of different modules in AmortizedPeriod. Specifically, we consider removing the trend, outliers, and the monthly basis in AmortizedPeriod. The results are presented in Table 2. Our findings indicate that it is crucial to consider the trend, outliers, and irregular periods during periodicity detection when such components exist in the data. As illustrated in Appendix G, Yahoo data are almost free of trends, outliers, and monthly periods, App Flow data is mainly corrupted by outliers, while the synthetic data consists of all these components. The results in Table 2 are consistent with these observations.

## 5 CONCLUSION

In this work, we have introduced AmortizedPeriod, a new model for periodicity detection that borrows the strength of both Bayesian statistics and deep learning. We have shown that AmortizedPeriod is more robust to trend changes, noise, outliers, missing data, and irregular periods than existing SOTA methods. The micro $F_1$-score resulting from AmortziedPeriod can surpass those of SOTA methods by $28\%$ on average. Moreover, the inference amortization technique excludes the need to conduct inference from scratch for every time series. Thus, the resulting run time is at least $55\%$ shorter than other methods. In addition, a small proportion of labeled time series can significantly boost the performance of AmortizedPeriod. We believe that this is the first self-supervised/semi-supervised model for periodicity identification that successfully solves the burning issues of limited robustness and memorylessness to the SOTA methods.

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

# A  SYMBOLS AND NOTATIONS

Table 3: Notations and their meanings.

| Notation | Size | Meaning |
|---|---|---|
| $q$ | Constant | exact period for a time series |
| $q(\cdot)$ | | variational $q$-distributions |
| $\boldsymbol{x}$ or $x(t)$ | $L \times 1$ | a time series with time stamp $t$ |
| $\boldsymbol{s}$ or $s(t)$ | $L \times 1$ | periodic/seasonal components with time stamp $t$ |
| $\mathcal{S}_q$ | | Ramanujan subspace with period $q$ |
| $\boldsymbol{R}_q$ | $L \times \phi(q)$ | Linear basis corresponding to the subspace $\mathcal{S}_q$ |
| $\boldsymbol{D}$ | $L \times \sum_{q=1}^{P_{\max}} \phi(q)$ | Ramanujan dictionary with a maximum period of $P_{\max}$ |
| $\boldsymbol{\alpha}$ | $\sum_{q=1}^{P_{\max}} \phi(q) \times 1$ | coefficients w.r.t. each basis in the Ramanujan dictionary |
| $\boldsymbol{\tau}$ or $\boldsymbol{\tau}(t)$ | $L \times 1$ | trend component |
| $\boldsymbol{\epsilon}$ or $\epsilon(t)$ | $L \times 1$ | noise component |
| $\boldsymbol{\delta}$ or $\delta(t)$ | $L \times 1$ | outlier component |
| $\nu$ | $1 \times 1$ | global shrinkage parameter of the horse-shoe prior |
| $\sigma_j$ | $1 \times 1$ | local shrinkage parameter of the horse-shoe prior for each periodic coefficient $\alpha_j$ |
| $a_{\omega_j}, b_{\omega_j}$ | $1 \times 1$ | shape parameters of the beta prior for $\omega_j$ |
| $\lambda$ | $1 \times 1$ | inverse global shrinkage parameter of the horse-shoe prior |
| $\omega_j$ | $1 \times 1$ | inverse local shrinkage parameter in the horse-shoe prior for each periodic coefficient $\alpha_j$ |
| $a_\lambda$ | $1 \times 1$ | shape parameter of the Gamma prior for $\lambda$ |
| $b_\lambda$ | $1 \times 1$ | rate parameter of the Gamma prior for $\lambda$ |
| $\Delta^2 \tau(t)$ | $1 \times 1$ | second order differences, that is, $\Delta^2 \tau(t) = (\tau(t+1) - \tau(t)) - (\tau(t) - \tau(t-1))$ |
| $\beta_1$ | $1 \times 1$ | smoothness parameter of the thin-plate prior that controls the curvature of the trend over time |
| $\beta_0$ | $1 \times 1$ | parameter of the thin-plate prior that controls how close to zero the trend is |
| $\boldsymbol{K}_{tp}$ | $L \times L$ | weighted Laplacian matrix of the thin-plate prior |
| $\boldsymbol{I}$ | $L \times L$ | identity matrix |
| $a_{\beta_0}$ | $1 \times 1$ | shape parameter of the Gamma prior for $\beta_0$ |
| $b_{\beta_0}$ | $1 \times 1$ | rate parameter of the Gamma prior for $\beta_0$ |
| $a_{\beta_1}$ | $1 \times 1$ | shape parameter of the Gamma prior for $\beta_1$ |
| $b_{\beta_1}$ | $1 \times 1$ | rate parameter of the Gamma prior for $\beta_1$ |
| $v_{\delta(t)}$ | $1 \times 1$ | variance of the zero-mean Gaussian prior for the outlier component $\delta(t)$ at timestamp $t$ |
| $a_{\delta(t)}$ | $1 \times 1$ | shape parameter of the inverse Gamma prior for $v_{\delta(t)}$ |
| $b_{\delta(t)}$ | $1 \times 1$ | scale parameter of the inverse Gamma prior for $v_{\delta(t)}$ |
| $v_\epsilon$ | $1 \times 1$ | variance of the zero-mean Gaussian prior for the noise component across all timestamps |
| $\boldsymbol{x}^{\{i\}}$ | $L \times 1$ | the $i$-th time series, where the superscript denotes the index |
| $\boldsymbol{K}^{\{i\}}$ | $(\sum_{q=1}^{P_{\max}} \phi(q) + L) \times (\sum_{q=1}^{P_{\max}} \phi(q) + L)$ | the precision matrix of the Gaussian variational distribution $q(\boldsymbol{\alpha}^{\{i\}}, \boldsymbol{\tau}^{\{i\}} \| \boldsymbol{x}^{\{i\}}, \boldsymbol{\omega}, \lambda, \beta_0, \beta_1, v_\epsilon^{\{i\}}, \boldsymbol{v}_{\boldsymbol{\delta}}^{\{i\}})$ |
| $\boldsymbol{h}^{\{i\}}$ | $(\sum_{q=1}^{P_{\max}} \phi(q) + L) \times 1$ | the potential vector of the Gaussian variational distribution $q(\boldsymbol{\alpha}^{\{i\}}, \boldsymbol{\tau}^{\{i\}} \| \boldsymbol{x}^{\{i\}}, \boldsymbol{\omega}, \lambda, \beta_0, \beta_1, v_\epsilon^{\{i\}}, \boldsymbol{v}_{\boldsymbol{\delta}}^{\{i\}})$ |
| $m_{\delta(t)}^{\{i\}}(\boldsymbol{x}^{\{i\}})$ | $1 \times 1$ | the mean of the log-normal variational distribution for $v_{\delta(t)}^{\{i\}}$ $q(v_{\delta(t)}^{\{i\}} \| \boldsymbol{x}^{\{i\}})$, which is a function of $\boldsymbol{x}^{\{i\}}$ determined by the inference network |
| $\nu_{\delta(t)}^{\{i\}}(\boldsymbol{x}^{\{i\}})$ | $1 \times 1$ | the variance of the log-normal variational distribution for $v_{\delta(t)}^{\{i\}}$ $q(v_{\delta(t)}^{\{i\}} \| \boldsymbol{x}^{\{i\}})$, which is a function of $\boldsymbol{x}^{\{i\}}$ determined by the inference network |
| $m_\epsilon^{\{i\}}(\boldsymbol{x}^{\{i\}})$ | $1 \times 1$ | the mean of the log-normal variational distribution for $v_\epsilon^{\{i\}}$ $q(v_\epsilon^{\{i\}} \| \boldsymbol{x}^{\{i\}})$, which is a function of $\boldsymbol{x}^{\{i\}}$ determined by the inference network |
| $\nu_\epsilon^{\{i\}}(\boldsymbol{x}^{\{i\}})$ | $1 \times 1$ | the variance of the log-normal variational distribution for $v_\epsilon^{\{i\}}$ $q(v_\epsilon^{\{i\}} \| \boldsymbol{x}^{\{i\}})$, which is a function of $\boldsymbol{x}^{\{i\}}$ determined by the inference network |
| $c_{\omega_j}(\boldsymbol{x}^{\{1:N\}}), d_{\omega_j}(\boldsymbol{x}^{\{1:N\}})$ | $1 \times 1$ | the shape parameters of the Kumaraswamy distribution for $\omega_j$ $q(\omega_j \| \boldsymbol{x}^{\{1:N\}})$, , which is a function of $\boldsymbol{x}^{\{1:N\}}$ determined by the inference network |
| $m_\lambda(\boldsymbol{x}^{\{1:N\}})$ | $1 \times 1$ | the mean of the log-normal variational distribution for $\lambda$ $q(\lambda \| \boldsymbol{x}^{\{1:N\}})$, which is a function of $\boldsymbol{x}^{\{1:N\}}$ determined by the inference network |
| $\nu_\lambda(\boldsymbol{x}^{\{1:N\}})$ | $1 \times 1$ | the variance of the log-normal variational distribution for $\lambda$ $q(\lambda \| \boldsymbol{x}^{\{1:N\}})$, which is a function of $\boldsymbol{x}^{\{1:N\}}$ determined by the inference network |
| $m_{\beta_0}(\boldsymbol{x}^{\{1:N\}})$ | $1 \times 1$ | the mean of the log-normal variational distribution for $\beta_0$ $q(\beta_0 \| \boldsymbol{x}^{\{1:N\}})$, which is a function of $\boldsymbol{x}^{\{1:N\}}$ determined by the inference network |
| $\nu_{\beta_0}(\boldsymbol{x}^{\{1:N\}})$ | $1 \times 1$ | the variance of the log-normal variational distribution for $\beta_0$ $q(\beta_0 \| \boldsymbol{x}^{\{1:N\}})$, which is a function of $\boldsymbol{x}^{\{1:N\}}$ determined by the inference network |
| $m_{\beta_1}(\boldsymbol{x}^{\{1:N\}})$ | $1 \times 1$ | the mean of the log-normal variational distribution for $\beta_1$ $q(\beta_1 \| \boldsymbol{x}^{\{1:N\}})$, which is a function of $\boldsymbol{x}^{\{1:N\}}$ determined by the inference network |
| $\nu_{\beta_1}(\boldsymbol{x}^{\{1:N\}})$ | $1 \times 1$ | the variance of the log-normal variational distribution for $\beta_1$ $q(\beta_1 \| \boldsymbol{x}^{\{1:N\}})$, which is a function of $\boldsymbol{x}^{\{1:N\}}$ determined by the inference network |
| $\mathbb{E}_q[f]$ | | expectation of a function $f$ over a distribution $q$ |
| $D_{\mathrm{KL}}(q \| p)$ | | KL divergence between distributions $q$ and $p$ |
| $\mathcal{L}$ | | the ELBO |
| $\mathbb{H}_q$ | | the entropy of distribution $q$ |
| $\mathbf{A}, \mathbf{B}$ | | input tensors to the cross attention block (CAB) |
| $\mathbf{Q}, \mathbf{K}, \mathbf{V}$ | | query, key, and value tensors as the input of the multihead attention (MHA) |
| MHA | | multihead attention |
| FFN | | feedforward network |
| LN | | layer norm |
| CAB | | cross attention block |
| SAB | | self attention block |

# B  MORE DETAILS ON RELATED WORKS

Table 4 presents a comprehensive comparison of the proposed AmortizedPeriod with existing methods discussed in Section 2, based on the six essential criteria: 1) capability to handle multiple periods, 2) ability to accommodate irregular periods, 3) consideration of trend, 4) robustness to noise, 5) handling

of outliers, and 6) handling of missing data. Analysis of the table reveals that the existing methods address only a subset of these requirements, whereas AmortizedPeriod demonstrates robustness by effectively addressing all of them.

Table 4: Comparision between AmortizedPeriod and the existing works in the four groups for periodicity identification in terms of robustness.

| Group | Method | Single Period | Multiple Periods | Irregular Periods | Trends | Noise | Outliers | Missing Data |
|---|---|---|---|---|---|---|---|---|
| 1. ACF | Elfeky et al. (2004) | Yes | Yes | No | No | Yes | No | No |
| | Wang et al. (2006) | Yes | No | No | Yes | Yes | No | No |
| | Toller & Kern (2017) | Yes | No | No | Yes | Yes | No | Yes |
| | Breitenbach et al. (2023) | Yes | No | No | Yes | Yes | No | No |
| 2. DFT | Tominaga (2010) | Yes | Yes | No | No | Yes | No | No |
| | Drutsa et al. (2017) | Yes | Yes | No | No | Yes | No | No |
| | Bauer et al. (2020) | Yes | Yes | No | Yes | Yes | No | No |
| 3. DFT& ACF | AutoPeriod (Vlachos et al., 2005) | Yes | Yes | No | No | Yes | No | No |
| | Sazed (Toller et al., 2019) | Yes | No | No | Yes | Yes | No | No |
| | Puech et al. (2020) | Yes | Yes | No | Yes | Yes | No | No |
| | RobustPeriod (Wen et al., 2021) | Yes | Yes | No | Yes | Yes | Yes | No |
| | Wen et al. (2023) | Yes | Yes | No | Yes | Yes | Yes | Yes |
| 4. RPD | Tenneti & Vaidyanathan (2015; 2016) | Yes | Yes | No | No | Yes | No | No |
| | Zhang et al. (2020) | Yes | Yes | No | No | Yes | No | Yes |
| | **AmortizedPeriod** | Yes | Yes | Yes | Yes | Yes | Yes | Yes |

## C   FORMAL DEFINITION OF THE RAMANUJAN SUBSPACES

The Ramanujan Subspaces can be defined from two distinct perspectives, namely the Fourier basis and the Ramanujan sum.

From the Fourier viewpoint, the $q$-th Ramanujan subspace $\mathcal{S}_q$ encompasses all exactly $q$-periodic time series $\boldsymbol{s}_q(t)$ that cannot be further decomposed into smaller periods. It can be represented as:

$$\boldsymbol{s}_q(t) = \sum_{\substack{k=1, \\ (k,q)=1}}^{q} a_k W_q^{kt}, \tag{13}$$

where $W_q = \exp(j2\pi/q)$ is the Fourier basis, $a_k$ are the coefficient for the frequency $2\pi k/q$, and $(k, q) = 1$ means that the greatest common divisor (GCD) of $k$ and $q$ is 1. Note that $\mathcal{S}_q$ only contains non-zero coefficients at the "coprime frequencies" $2\pi k/q$, where $1 \leq k \leq q$ and $k$ is coprime to $q$, and therefore, the exact period of $\boldsymbol{s}(t)$ is $q$ and not smaller. It follows that the rank of this subspace $\mathcal{S}_q$ is $\phi(q)$, which is the Euler totient function signifying the number of integers $k$ in $1 \leq k \leq q$ satisfying $(k, q) = 1$ (Vaidyanathan, 2014a).

On the other hand, the renowned Indian mathematician Srinivasa Ramanujan introduced the concept of the Ramanujan sum in 1918 (Ramanujan, 1918), which is given by

$$r_q(t) = \sum_{\substack{k=1, \\ (k,q)=1}}^{q} W_q^{kt}. \tag{14}$$

Ramanujan employed this sum to demonstrate that various standard arithmetic functions in number theory can be expressed as linear combinations of $r_q(t)$. For any fixed integer $q$, $r_q(t)$ is a sequence with periodicity $q$ that cannot be further decomposed. Interestingly, different from the complex-valued Fourier basis, the Ramanujan sum is always integer-valued, which is an appealing characteristic. Moreover, it can be easily obtained via efficient recursive computations, as demonstrated in previous works (Vaidyanathan, 2014a).

By leveraging the definition of the Ramanujan sum, the $q$-th Ramanujan subspace can also be spanned by all the $\phi(q)$ circularly shifted versions of $r_q(t)$ (Vaidyanathan, 2014b):

$$s_q(t) = \sum_{k=0}^{\phi(q)-1} b_k r_q(t-k). \tag{15}$$

where $b_k$ denotes the real number coefficients. Since the rank of $\mathcal{S}_q$ is $\phi(q)$ as previously mentioned, any set of $\phi(q)$ linearly independent vectors from $\mathcal{S}_q$ can span the entire subspace. The $\phi(q)$ circularly shifted versions of $r_q(t)$ serve as a set of such linearly independent vectors.

Consider the sum $\boldsymbol{x}(t) = \sum_i \boldsymbol{x}_{q_i}(t)$, where $\boldsymbol{x}_{q_i}(t)$ is an arbitrary $q_i$-period time series (cf. Definition 1), the period of $\boldsymbol{x}(t)$ is equal to the least common multiple (LCM) of $q_i$, i.e., $\text{LCM}(q_i)$, or a divisor of $\text{LCM}(q_i)$, as proven in (Vaidyanathan, 2014a; Tenneti & Vaidyanathan, 2015). For instance, given two 2-periodic time series $\{1, -1, 1, -1, \cdots\}$ and $\{-1, 1, -1, 1, \cdots\}$, the period of their sum is 1, which is a divisor of $\text{LCM}(2, 2) = 2$. However, if the exact period of $\boldsymbol{x}_{q_i}(t)$ is $q_i$, that is, $\boldsymbol{x}_{q_i}(t) \in \mathcal{S}_{q_i}$, the period of $\boldsymbol{x}(t)$ can only be $\text{LCM}(q_i)$ and not smaller. This is another attractive property of the Ramanujan subspace. Conversely, any arbitrary $q$-periodic time series, as defined in Definition 1, can be spanned by the set of Ramanujan subspaces $\mathcal{S}_{q_i}$, where $q_i$ represents all divisors of $q$, including both 1 and $q$. For example, a 6-periodic time series can be obtained by adding periodic components from $\mathcal{S}_2$ and $\mathcal{S}_3$, $\mathcal{S}_1$ and $\mathcal{S}_6$, or $\mathcal{S}_1$, $\mathcal{S}_2$, $\mathcal{S}_3$, and $\mathcal{S}_6$, as long as the LCM of the exact periods corresponding to these subspaces equal 6. In other words, we can summarize this concept with the following proposition:

**Proposition 2.** *According to the definition of periods in Definition 1 and exact periods in Definition 2, the period of a time series is equal to the least common multiple (LCM) of its exact periods.*

In order to determine the fundamental exact periods of a time series $\boldsymbol{x}(t)$, we can decompose $\boldsymbol{x}(t)$ as

$$\boldsymbol{x} = \boldsymbol{D}\boldsymbol{\alpha}, \tag{16}$$

where $\boldsymbol{D} = [\boldsymbol{R}_1, \boldsymbol{R}_2, \cdots, \boldsymbol{R}_{P_{\max}}]$ denotes the Ramanujan dictionary, and $\boldsymbol{R}_q$ are the matrices of size $L \times \phi(q)$ whose columns are the $\phi(q)$ shifted versions of $r_q(t)$ (14). Ideally, the coefficient vector $\boldsymbol{\alpha}$ has non-zero entries only at locations where the periodic components of $\boldsymbol{x}$ reside. The number of columns in $\boldsymbol{D}$ is $\Phi(P_{\max}) = \sum_{q=1}^{P_{\max}} \phi(q)$, that is, $\mathcal{O}(P_{\max}^2)$. Hence, $\boldsymbol{D}$ can be a fat matrix with more columns than rows. The resulting problem of estimating $\boldsymbol{\alpha}$ given $\boldsymbol{x}$ and $\boldsymbol{D}$ can be formulated as a sparse vector recovery problem such as basis pursuit or lasso (Tenneti & Vaidyanathan, 2016).

## D  A Brief Introduction to the Priors for AmortizedPeriod

### D.1  Student's $t$-priors

Recall that we use student's $t$-priors to describe the behavior of outliers. To remain meaningful, the outliers must be relatively rare. This means that the vector $\boldsymbol{\delta}$ should be sparse, with most of its entries being zero. Assuming that the components of $\boldsymbol{\delta}$ are independent and identically distributed random variables, each entry $\boldsymbol{\delta}(t)$ of the outlier vector may follow a normal distribution with zero mean and variance $v_{\delta(t)}$, as represented by the equation:

$$p(\boldsymbol{\delta}(t)|v_{\delta(t)}) = \mathcal{N}(\boldsymbol{\delta}(t); 0, v_{\delta(t)}), \tag{17}$$

The outlier precision $v_{\delta(t)}$ is also a random variable, and since we have no prior knowledge of its value, we adopt a non-informative conjugate prior:

$$p(v_{\delta(t)}) = \text{Inv-Gamma}(v_{\delta(t)}; a_{\delta(t)}, b_{\delta(t)}). \tag{18}$$

Here, $\text{Inv-Gamma}(v_{\delta(t)}; a_{\delta(t)}, b_{\delta(t)})$ refers to the inverse Gamma distribution with shape parameter $a_{\delta(t)}$ and scale parameter $b_{\delta(t)}$. By setting both $a_{\delta(t)}$ and $b_{\delta(t)}$ to small values, such as $10^{-4}$, the prior becomes non-informative.

As a result, the marginal distribution for each outlier can be obtained by integrating out $v_{\delta(t)}$, yielding (Yu et al., 2020a):

$$p(\boldsymbol{\delta}(t)) = \int_0^\infty p(\boldsymbol{\delta}(t)|v_{\delta(t)}) p(v_{\delta(t)}) \, \mathrm{d}v_{\delta(t)} \tag{19}$$

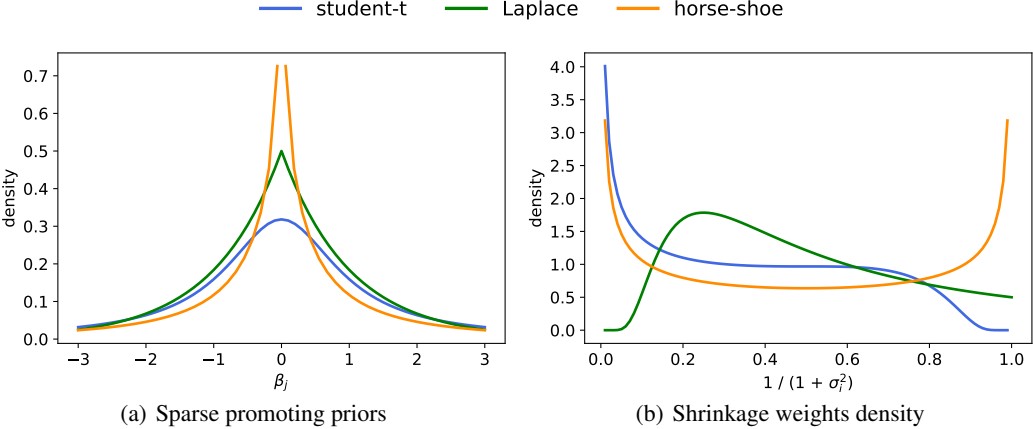

(a) Sparse promoting priors        (b) Shrinkage weights density

Figure 4: The density of the commonly used sparse promoting priors, including the Student's T, Laplace, and horse-shoe prior (a), and the density of the corresponding shrinkage weights $1/(1+\sigma^2)$ (b).

$$= \frac{\Gamma(a_{\delta(t)} + \frac{1}{2})}{\Gamma(a_{\delta(t)})\sqrt{2\pi b_{\delta(t)}}} \left( \frac{1}{1 + \frac{\delta_{ij}^2}{2b_{\delta(t)}}} \right)^{a_{\delta(t)} + \frac{1}{2}}. \tag{20}$$

This equation represents a Student's $t$-distribution. On one hand, the Student's $t$-distribution tends to shrink most entries in $\delta$ towards zero due to its sharp peak at zero. On the other hand, the long tail of the distribution allows for some elements $\delta(t)$ to be far from zero. Consequently, the resulting vector $\delta$ will be sparse.

### D.2    HORSE-SHOE PRIORS

The horse-shoe prior is also a Bayesian sparsity-promoting prior that assumes that each coefficient $\alpha_i$ can be expressed as a scale mixture of Gaussians (Carvalho et al., 2009; Yu et al., 2020b):

$$p(\alpha_j|\sigma_j, \nu) = \mathcal{N}(\alpha_j; 0, \nu^2\sigma_j^2), \tag{21}$$

$$p(\sigma_j) = C^+(0, 1), \tag{22}$$

where the standard deviation $\sigma_j$ follows a half-Cauchy distribution $C^+(0, 1)$. Here, $\nu$ is referred to as the global shrinkage parameter, and $\sigma_j$ as the local shrinkage parameter.

Figure 4(a) displays the densities for commonly-used sparsity-promoting priors, including the horse-shoe, Laplacian, and Student's $t$ priors. The horse-shoe prior is particularly useful as a shrinkage prior for sparse Bayesian learning due to its unique characteristics. Its flat, Cauchy-like tails allow the coefficients of the strong periods to remain large, while its infinitely tall spike at the origin provides severe shrinkage for the zero elements in $\alpha$. This property is not present in the Laplacian and $t$ prior.

Viewed another way, we can represent the shrinkage weight $\omega_j$ as $\omega_j = 1/(1+\sigma_j^2)$, and depict its density for the horse-shoe, Laplacian, and $t$ prior in Figure 4(b). Choosing $\sigma_j \sim C^+(0, 1)$ implies $\omega_j \sim \text{Beta}(0.5, 0.5)$, which is a symmetric density unbounded at both 0 and 1. In other words, the horse-shoe prior yields a shrinkage profile that exhibits two expected outcomes: large nonzero values ($\omega_j \approx 0$, implying little shrinkage), and zeros ($\omega_j \approx 1$, indicating total shrinkage). In contrast, the Laplacian prior leads to a shrinkage profile that approaches a fixed constant as $\omega_j$ approaches 1, while vanishing entirely as $\omega_j$ approaches 0. The $t$ prior exhibits an unbounded density near $\omega_j = 0$, while being bounded near $\omega_j = 1$.

In Section 3.2, we further reparameterize the horse-shoe prior using $\omega_j$ and $\lambda = 1/\nu^2$ to utilize $\omega_j$ as an indicator of which periods exist.

## D.3 THIN-PLATE PRIORS

The thin-plate model (Rue & Held, 2005; Yu & Dauwels, 2016), commonly used as a smoothness prior, is a Gaussian Markov random field (GMRF) that penalizes second-order differences. Specifically, the second-order differences are modeled as a Gaussian distribution:

$$\Delta^2 \boldsymbol{\tau}(t) \sim \mathcal{N}(0, \beta_1^{-1}). \tag{23}$$

For a one-dimensional problem with variables $\boldsymbol{\tau}(t)$ evenly distributed along a chain, the second-order difference at $\boldsymbol{\tau}(t)$ can be defined as $\Delta^2 \boldsymbol{\tau}(t) = \boldsymbol{\tau}(t-1) - 2\boldsymbol{\tau}(t) + \boldsymbol{\tau}(t+1)$. Consequently, the density function of a thin-plate model with a chain structure can be expressed as:

$$p(\boldsymbol{\tau}) \propto \exp\left(-\frac{\beta_1}{2} \sum_{t=2}^{L-1} \left(\boldsymbol{\tau}(t-1) - 2\boldsymbol{\tau}(t) + \boldsymbol{\tau}(t+1)\right)^2\right) \propto \exp\left(-\frac{\beta_1}{2} \boldsymbol{\tau}^T \boldsymbol{K}_{tp} \boldsymbol{\tau}\right), \tag{24}$$

where the smoothness parameter $\beta_1$ controls the curvature, and $\boldsymbol{K}_{tp}$ takes the form:

$$\boldsymbol{K}_{tp} = A^T A, \tag{25}$$

$$A = \begin{bmatrix} 1 & -2 & 1 & & \\ & \ddots & \ddots & \ddots & \\ & & 1 & -2 & 1 \end{bmatrix}. \tag{26}$$

Note that $A$ is a $(L-2) \times L$ matrix. Apparently, the precision matrix (i.e., inverse covariance matrix) of a thin-plate model is given by $K = \alpha_z K_{tp}$. From equation (24), it is evident that the thin-plate model remains unchanged by the addition of a constant or a linear function along the Markov chain. In other words, $\boldsymbol{K}_{tp}\boldsymbol{1} = \boldsymbol{0}$ and $\boldsymbol{K}_{tp}\boldsymbol{s}_1 = \boldsymbol{0}$, where $\boldsymbol{1}$ is a column vector of ones, and $\boldsymbol{s}_1 = [1, 2, ..., P]^T$. As a result, this prior can accommodate linear trends without any penalty. Additionally, it can be inferred that $K_{tp}$ has two zero eigenvalues, making it rank deficient. Consequently, an improper density is often employed in practice (Rue & Held, 2005), given by:

$$p(\boldsymbol{\tau}) \propto |K|_+^{0.5} \exp\left\{-\frac{1}{2}\boldsymbol{\tau}^T K \boldsymbol{\tau}\right\}, \tag{27}$$

where $|K|_+$ denotes the product of the positive eigenvalues of the precision matrix $K$. Eq. (25) eveals that the conditional mean of one variable $\boldsymbol{\tau}(t)$, conditioned on the remaining variables $\boldsymbol{\tau}(1, \cdots, t-1, t+1, \cdots, L)$, can be written as (Rue & Held, 2005):

$$E(\boldsymbol{\tau}(t)|\boldsymbol{\tau}(1, \cdots, t-1, t+1, \cdots, L)) = \frac{4}{6}(\boldsymbol{\tau}(t+1) + \boldsymbol{\tau}(t-1)) - \frac{1}{6}(\boldsymbol{\tau}(t+2) + \boldsymbol{\tau}(t-2)). \tag{28}$$

This can be seen as second-order polynomial interpolation based on four nearby variables $+\boldsymbol{\tau}(t-2)$, $\boldsymbol{\tau}(t-1)$, $\boldsymbol{\tau}(t+1)$, and $\boldsymbol{\tau}(t+2)$ without an overall level. Therefore, the thin-plate model allows for deviations from an overall mean level without requiring the specification of the overall mean level itself. This property is often desirable in practice.

# E   PROOF OF PROPOSITION 1

As mentioned in Section 3.3, we set $q(\boldsymbol{\alpha}^{\{i\}}, \boldsymbol{\tau}^{\{i\}}|\boldsymbol{x}^{\{i\}}, \boldsymbol{\omega}, \lambda, \beta_0, \beta_1, v_\epsilon^{\{i\}}, \boldsymbol{v}_\delta^{\{i\}}) = p(\boldsymbol{\alpha}^{\{i\}}, \boldsymbol{\tau}^{\{i\}}|\boldsymbol{x}^{\{i\}}, \boldsymbol{\omega}, \lambda, \beta_0, \beta_1, v_\epsilon^{\{i\}}, \boldsymbol{v}_\delta^{\{i\}})$, in order to capture the dependencies between $\boldsymbol{\alpha}^{\{i\}}$ and $\boldsymbol{\tau}^{\{i\}}$ given the remaining latent variables. Recall that

$$p(\boldsymbol{x}^{\{i\}}|\boldsymbol{\alpha}^{\{i\}}, \boldsymbol{\tau}^{\{i\}}, v_\epsilon^{\{i\}}, \boldsymbol{v}_\delta^{\{i\}}) \propto \exp\left(-\frac{1}{2}(\boldsymbol{x} - \boldsymbol{D}\boldsymbol{\alpha} - \boldsymbol{\tau})^T (v_\epsilon^{-1}\boldsymbol{I} + \text{diag}(\boldsymbol{v}_\delta)^{-1})(\boldsymbol{x} - \boldsymbol{D}\boldsymbol{\alpha} - \boldsymbol{\tau})\right),$$

$$p(\alpha_j^{\{i\}}|\omega_j, \lambda) \propto \exp\left(-\frac{\lambda}{2}\frac{1-\omega_j}{\omega_j}\alpha_j^{\{i\}2}\right),$$

$$p(\boldsymbol{\tau}^{\{i\}}|\beta_0, \beta_1) \propto \exp\left(-\frac{1}{2}\boldsymbol{\tau}^T(\beta_0\boldsymbol{I} + \beta_1\boldsymbol{K}_{tp})\boldsymbol{\tau}\right).$$

All these distributions are Gaussian. According to the properties of Gaussian distributions (Lin et al., 2018), we have

$$p(\boldsymbol{\alpha}^{\{i\}}, \boldsymbol{\tau}^{\{i\}} | \boldsymbol{x}^{\{i\}}, \boldsymbol{\omega}, \lambda, \beta_0, \beta_1, v_\epsilon^{\{i\}}, \boldsymbol{v}_{\boldsymbol{\delta}}^{\{i\}})$$

$$\propto p(\boldsymbol{x}^{\{i\}} | \boldsymbol{\alpha}^{\{i\}}, \boldsymbol{\tau}^{\{i\}}, v_\epsilon^{\{i\}}, \boldsymbol{v}_{\boldsymbol{\delta}}^{\{i\}}) \prod_{j=1}^{\Phi} \left[ p(\alpha_j^{\{i\}} | \omega_j, \lambda) \right] p(\boldsymbol{\tau}^{\{i\}} | \beta_0, \beta_1)$$

$$\propto \exp\left( -\frac{1}{2} \begin{bmatrix} \boldsymbol{\alpha}^{\{i\}^T} & \boldsymbol{\tau}^{\{i\}^T} \end{bmatrix} \boldsymbol{K}^{\{i\}} \begin{bmatrix} \boldsymbol{\alpha}^{\{i\}} \\ \boldsymbol{\tau}^{\{i\}} \end{bmatrix} + \begin{bmatrix} \boldsymbol{\alpha}^{\{i\}^T} & \boldsymbol{\tau}^{\{i\}^T} \end{bmatrix} \boldsymbol{h}^{\{i\}} \right)$$

$$= \mathcal{N}\left( \begin{bmatrix} \boldsymbol{\alpha}^{\{i\}} \\ \boldsymbol{\tau}^{\{i\}} \end{bmatrix}; \boldsymbol{K}^{\{i\}^{-1}} \boldsymbol{h}^{\{i\}}, \boldsymbol{K}^{\{i\}^{-1}} \right), \tag{29}$$

where

$$\boldsymbol{K}^{\{i\}} = \begin{bmatrix} \lambda \operatorname{diag}\left(\frac{1-\omega_j}{\omega_j}\right) + \boldsymbol{D}^T \operatorname{diag}\left(v_\epsilon^{\{i\}} + \boldsymbol{v}_{\boldsymbol{\delta}}^{\{i\}}\right)^{-1} \boldsymbol{D} & \boldsymbol{D}^T \operatorname{diag}\left(v_\epsilon^{\{i\}} + \boldsymbol{v}_{\boldsymbol{\delta}}^{\{i\}}\right)^{-1} \\ \operatorname{diag}\left(v_\epsilon^{\{i\}} + \boldsymbol{v}_{\boldsymbol{\delta}}^{\{i\}}\right)^{-1} \boldsymbol{D} & \beta_0 \boldsymbol{I} + \beta_1 \boldsymbol{K}_{tp} + \operatorname{diag}\left(v_\epsilon^{\{i\}} + \boldsymbol{v}_{\boldsymbol{\delta}}^{\{i\}}\right)^{-1} \end{bmatrix}, \tag{30}$$

$$\boldsymbol{h}^{\{i\}} = \begin{bmatrix} \boldsymbol{D}^T \\ \boldsymbol{I} \end{bmatrix} \operatorname{diag}\left(v_\epsilon^{\{i\}} + \boldsymbol{v}_{\boldsymbol{\delta}}^{\{i\}}\right)^{-1} \boldsymbol{x}^{\{i\}}. \tag{31}$$

In addition, when there exist missing data, the time points $t$ in the time series $\boldsymbol{x}^{\{i\}}(1:L)$ can be partitioned into two sets $\mathbb{M}$ and $\mathbb{O}$, representing the missing and observed set respectively. The posterior distribution of $\boldsymbol{\alpha}^{\{i\}}$ and $\boldsymbol{\tau}^{\{i\}}$ can be computed in a natural way as:

$$p(\boldsymbol{\alpha}^{\{i\}}, \boldsymbol{\tau}^{\{i\}} | \boldsymbol{x}_{\mathbb{O}}^{\{i\}}, \boldsymbol{\omega}, \lambda, \beta_0, \beta_1, v_\epsilon^{\{i\}}, \boldsymbol{v}_{\boldsymbol{\delta}}^{\{i\}})$$

$$\propto \prod_{t \in \mathbb{O}} p(\boldsymbol{x}^{\{i\}}(t) | \boldsymbol{\alpha}^{\{i\}}(t), \boldsymbol{\tau}^{\{i\}}(t), v_\epsilon^{\{i\}}, \boldsymbol{v}_{\boldsymbol{\delta}(t)}^{\{i\}}) \prod_{j=1}^{\Phi} \left[ p(\alpha_j^{\{i\}} | \omega_j, \lambda, \right] p(\boldsymbol{\tau}^{\{i\}} | \beta_0, \beta_1). \tag{32}$$

The above distribution is still Gaussian, and the corresponding $\boldsymbol{K}^{\{i\}}$ and $\boldsymbol{h}^{\{i\}}$ can be computed by replacing $\boldsymbol{x}^{\{i\}}(t)$ for $t \in \mathbb{M}$ with 0 in (30) and (31).

## F  DERIVATION OF THE ELBO

Recall that the generative model (i.e., the $p$ distribution) can be factorized as:

$$p(\boldsymbol{x}^{\{1:N\}}, \boldsymbol{\alpha}^{\{1:N\}}, \boldsymbol{\omega}, \lambda, \boldsymbol{\tau}^{\{1:N\}}, \beta_0, \beta_1, v_\epsilon^{\{1:N\}}, \boldsymbol{v}_{\boldsymbol{\delta}}^{\{1:N\}})$$

$$= \prod_{i=1}^{N} \left[ p(\boldsymbol{x}^{\{i\}} | \boldsymbol{\alpha}^{\{i\}}, \boldsymbol{\tau}^{\{i\}}, v_\epsilon^{\{i\}}, \boldsymbol{v}_{\boldsymbol{\delta}}^{\{i\}}) p(\boldsymbol{\alpha}^{\{i\}} | \boldsymbol{\omega}, \lambda) p(\boldsymbol{\tau}^{\{i\}} | \beta_0, \beta_1) p(v_\epsilon^{\{i\}}) p(\boldsymbol{v}_{\boldsymbol{\delta}}^{\{i\}}) \right]$$

$$\cdot \prod_{j=1}^{\Phi} p(\omega_j) p(\lambda) p(\beta_0) p(\beta_1), \tag{33}$$

where

$$p(\boldsymbol{x}^{\{i\}} | \boldsymbol{\alpha}^{\{i\}}, \boldsymbol{\tau}^{\{i\}}, v_\epsilon^{\{i\}}, \boldsymbol{v}_{\boldsymbol{\delta}}^{\{i\}}) = \mathcal{N}\left( \boldsymbol{x}^{\{i\}}; \boldsymbol{D}\boldsymbol{\alpha}^{\{i\}} + \boldsymbol{\tau}^{\{i\}}, v_\epsilon^{\{i\}} \boldsymbol{I} + \operatorname{diag}(\boldsymbol{v}_{\boldsymbol{\delta}}^{\{i\}}) \right), \tag{34}$$

$$p(\boldsymbol{\alpha}_j^{\{i\}} | \omega_j, \lambda) = \mathcal{N}\left( \alpha_j; 0, \left( \lambda \frac{1-\omega_j}{\omega_j} \right)^{-1} \right), \tag{35}$$

$$p(\boldsymbol{\tau}^{\{i\}} | \beta_0, \beta_1) = \mathcal{N}\left( \boldsymbol{\tau}^{\{i\}}; \boldsymbol{0}, \left( \beta_0 \boldsymbol{I} + \beta_1 \boldsymbol{K}_{tp} \right)^{-1} \right), \tag{36}$$

$$p(v_\epsilon^{\{i\}}) = \text{Inv-Gamma}\left( v_\epsilon^{\{i\}}; a_\epsilon^{\{i\}}, b_\epsilon^{\{i\}} \right), \tag{37}$$

$$p(v_{\delta(t)}^{\{i\}}) = \text{Inv-Gamma}\left( v_{\delta(t)}^{\{i\}}; a_{\delta(t)}^{\{i\}}, b_{\delta(t)}^{\{i\}} \right), \tag{38}$$

$$p(\omega_j) = \mathrm{Beta}\left(\omega_j; a_{\omega j}, b_{\omega j}\right), \tag{39}$$

$$p(\lambda) = \mathrm{Gamma}\left(\lambda; a_\lambda, b_\lambda\right), \tag{40}$$

$$p(\beta_0) = \mathrm{Gamma}\left(\beta_0; a_{\beta_0}, b_{\beta_0}\right), \tag{41}$$

$$p(\beta_1) = \mathrm{Gamma}\left(\beta_1; a_{\beta_1}, b_{\beta_1}\right), \tag{42}$$

$$\tag{43}$$

and the inference model (i.e., the $q$ distribution) can be factorized as:

$$q(\boldsymbol{\alpha}^{\{1:N\}}, \boldsymbol{\omega}, \lambda, \boldsymbol{\tau}^{\{1:N\}}, \beta_0, \beta_1, v_\epsilon^{\{1:N\}}, \boldsymbol{v_\delta}^{\{1:N\}} | \boldsymbol{x}^{\{1:N\}})$$

$$= \prod_{i=1}^{N} \left[ q(\boldsymbol{\alpha}^{\{i\}}, \boldsymbol{\tau}^{\{i\}} | \boldsymbol{x}^{\{i\}}, \boldsymbol{\omega}, \lambda, \beta_0, \beta_1, v_\epsilon^{\{i\}}, \boldsymbol{v_\delta}^{\{i\}}) q(v_\epsilon^{\{i\}} | \boldsymbol{x}^{\{i\}}) q(\boldsymbol{v_\delta}^{\{i\}} | \boldsymbol{x}^{\{i\}}) \right]$$

$$\cdot \prod_{j=1}^{\Phi} q(\omega_j | \boldsymbol{x}^{\{1:N\}}) q(\lambda | \boldsymbol{x}^{\{1:N\}}) q(\beta_0 | \boldsymbol{x}^{\{1:N\}}) q(\beta_1 | \boldsymbol{x}^{\{1:N\}}), \tag{44}$$

where

$$q(\boldsymbol{\alpha}^{\{i\}}, \boldsymbol{\tau}^{\{i\}} | \boldsymbol{x}^{\{i\}}, \boldsymbol{\omega}, \lambda, \beta_0, \beta_1, v_\epsilon^{\{i\}}, \boldsymbol{v_\delta}^{\{i\}}) = \mathcal{N}\left( \begin{bmatrix} \boldsymbol{\alpha}^{\{i\}} \\ \boldsymbol{\tau}^{\{i\}} \end{bmatrix}; \boldsymbol{K}^{\{i\}^{-1}} \boldsymbol{h}^{\{i\}}, \boldsymbol{K}^{\{i\}^{-1}} \right), \tag{45}$$

$$q(v_\epsilon^{\{i\}} | \boldsymbol{x}^{\{i\}}) = \mathrm{Lognormal}\left( v_\epsilon^{\{i\}}; m_\epsilon^{\{i\}}(\boldsymbol{x}^{\{i\}}), \nu_\epsilon^{\{i\}}(\boldsymbol{x}^{\{i\}}) \right), \tag{46}$$

$$q(v_{\delta(t)}^{\{i\}} | \boldsymbol{x}^{\{i\}}) = \mathrm{Lognormal}\left( v_{\delta(t)}^{\{i\}}; m_{\delta(t)}^{\{i\}}(\boldsymbol{x}^{\{i\}}), \nu_{\delta(t)}^{\{i\}}(\boldsymbol{x}^{\{i\}}) \right), \tag{47}$$

$$q(\omega_j | \boldsymbol{x}^{\{1:N\}}) = \mathrm{Kumaraswamy}\left( \omega_j; c_{\omega j}(\boldsymbol{x}^{\{1:N\}}), d_{\omega j}(\boldsymbol{x}^{\{1:N\}}) \right), \tag{48}$$

$$q(\lambda | \boldsymbol{x}^{\{1:N\}}) = \mathrm{Lognormal}\left( \lambda; m_\lambda(\boldsymbol{x}^{\{1:N\}}), \nu_\lambda(\boldsymbol{x}^{\{1:N\}}) \right), \tag{49}$$

$$q(\beta_0 | \boldsymbol{x}^{\{1:N\}}) = \mathrm{Lognormal}\left( \beta_0; m_{\beta_0}(\boldsymbol{x}^{\{1:N\}}), \nu_{\beta_0}(\boldsymbol{x}^{\{1:N\}}) \right), \tag{50}$$

$$q(\beta_1 | \boldsymbol{x}^{\{1:N\}}) = \mathrm{Lognormal}\left( \beta_1; m_{\beta_1}(\boldsymbol{x}^{\{1:N\}}), \nu_{\beta_1}(\boldsymbol{x}^{\{1:N\}}) \right). \tag{51}$$

Note that the parameters of $q(v_\epsilon^{\{i\}}|\boldsymbol{x}^{\{i\}})$, $q(v_{\delta(t)}^{\{i\}}|\boldsymbol{x}^{\{i\}})$, $q(\omega_j|\boldsymbol{x}^{\{1:N\}})$, $q(\lambda|\boldsymbol{x}^{\{1:N\}})$, $q(\beta_0|\boldsymbol{x}^{\{1:N\}})$, and $q(\beta_1|\boldsymbol{x}^{\{1:N\}})$ are explicit functions of $\boldsymbol{x}^{\{1:N\}}$, which is parameterized by the inference network.

By substituting the $p$ (33) and $q$ distributions (44) into the ELBO (9), we can obtain:

$$\mathcal{L} = \mathbb{E}_q \Bigg[ \sum_{i=1}^{N} \bigg( \log p(\boldsymbol{x}^{\{i\}} | \boldsymbol{\alpha}^{\{i\}}, \boldsymbol{\tau}^{\{i\}}, v_\epsilon^{\{i\}}, \boldsymbol{v_\delta}^{\{i\}}) + \log p(\boldsymbol{\alpha}^{\{i\}} | \boldsymbol{z}, \lambda_0, \lambda_1) + \log p(\boldsymbol{\tau}^{\{i\}} | \beta_0, \beta_1)$$

$$+ \log p(v_\epsilon^{\{i\}}) + \log p(\boldsymbol{v_\delta}^{\{i\}}) \bigg) + \sum_{j=1}^{\Phi} \log p(\omega_j) + \log p(\lambda) + + \log p(\beta_0) + \log p(\beta_1)$$

$$- \sum_{i=1}^{N} \bigg( \log q(\boldsymbol{\alpha}^{\{i\}}, \boldsymbol{\tau}^{\{i\}} | \boldsymbol{x}^{\{i\}}, \boldsymbol{z}, \lambda_0, \lambda_1, \beta_0, \beta_1, v_\epsilon^{\{i\}}, \boldsymbol{v_\delta}^{\{i\}}) + \log q(v_\epsilon^{\{i\}} | \boldsymbol{x}^{\{i\}})$$

$$+ \log q(\boldsymbol{v_\delta}^{\{i\}} | \boldsymbol{x}^{\{i\}}) \bigg) - \sum_{j=1}^{\Phi} \log q(\omega_j | \boldsymbol{x}^{\{1:N\}}) - \log q(\lambda | \boldsymbol{x}^{\{1:N\}}) - \log q(\beta_0 | \boldsymbol{x}^{\{1:N\}})$$

$$- \log q(\beta_1 | \boldsymbol{x}^{\{1:N\}}) \Bigg]$$

$$= \sum_{i=1}^{N} \bigg( \mathbb{E}_q \Big[ \log p(\boldsymbol{x}^{\{i\}} | \boldsymbol{\alpha}^{\{i\}}, \boldsymbol{\tau}^{\{i\}}, v_\epsilon^{\{i\}}, \boldsymbol{v_\delta}^{\{i\}}) + \log p(\boldsymbol{\alpha}^{\{i\}} | \boldsymbol{\omega}, \lambda) + \log p(\boldsymbol{\tau}^{\{i\}} | \beta_0, \beta_1)$$

$$
\begin{aligned}
&- \log q(\boldsymbol{\alpha}^{\{i\}}, \boldsymbol{\tau}^{\{i\}} | \boldsymbol{x}^{\{i\}}, \boldsymbol{\omega}, \lambda, \beta_0, \beta_1, v_\epsilon^{\{i\}}, \boldsymbol{v}_{\boldsymbol{\delta}}^{\{i\}}) \Big] \\
&- D_{\mathrm{KL}}\Big( q(v_\epsilon^{\{i\}} | \boldsymbol{x}^{\{i\}}) \| p(v_\epsilon^{\{i\}}) \Big) - D_{\mathrm{KL}}\Big( q(\boldsymbol{v}_{\boldsymbol{\delta}}^{\{i\}} | \boldsymbol{x}^{\{i\}}) \| p(\boldsymbol{v}_{\boldsymbol{\delta}}^{\{i\}}) \Big) \\
&- \sum_{j=1}^{\Phi} D_{\mathrm{KL}}\Big( q(\omega_j | \boldsymbol{x}^{\{1:N\}}) \| p(\omega_j) \Big) - D_{\mathrm{KL}}\Big( q(\lambda | \boldsymbol{x}^{\{1:N\}}) \| p(\lambda) \Big) \\
&- D_{\mathrm{KL}}\Big( q(\beta_0 | \boldsymbol{x}^{\{1:N\}}) \| p(\beta_0) \Big) - D_{\mathrm{KL}}\Big( q(\beta_1 | \boldsymbol{x}^{\{1:N\}}) \| p(\beta_1) \Big),
\end{aligned}
\tag{52}
$$

where $D_{\mathrm{KL}}$ denotes the KL (Kullback-Leibler) divergence between two distributions. We now delve into the expectations in the above expression.

It follows from Appendix E that the first term amounts to

$$
\begin{aligned}
\mathbb{E}_q \Big[ &\log p(\boldsymbol{x}^{\{i\}} | \boldsymbol{\alpha}^{\{i\}}, \boldsymbol{\tau}^{\{i\}}, v_\epsilon^{\{i\}}, \boldsymbol{v}_{\boldsymbol{\delta}}^{\{i\}}) + \log p(\boldsymbol{\alpha}^{\{i\}} | \boldsymbol{\omega}, \lambda) + \log p(\boldsymbol{\tau}^{\{i\}} | \beta_0, \beta_1) \\
&- \log q(\boldsymbol{\alpha}^{\{i\}}, \boldsymbol{\tau}^{\{i\}} | \boldsymbol{x}^{\{i\}}, \boldsymbol{\omega}, \lambda, \beta_0, \beta_1, v_\epsilon^{\{i\}}, \boldsymbol{v}_{\boldsymbol{\delta}}^{\{i\}}) \Big] \\
= &- \frac{1}{2} \sum_{t=1}^{L} \langle \log \left( v_\epsilon^{\{i\}} + v_{\delta(t)}^{\{i\}} \right) \rangle + \frac{1}{2} \sum_{j=1}^{\Phi} \langle \log \left( \lambda \frac{1 - \omega_j}{\omega_j} \right) \rangle \\
&+ \frac{1}{2} \langle \log \det(\beta_0 \boldsymbol{I} + \beta_1 \boldsymbol{K}_{tp}) \rangle - \frac{1}{2} \langle \log \det K \rangle - \frac{1}{2} \boldsymbol{x}^{\{i\}T} \langle \mathrm{diag}\left( v_\epsilon^{\{i\}} + \boldsymbol{v}_{\boldsymbol{\delta}}^{\{i\}} \right)^{-1} \rangle \boldsymbol{x}^{\{i\}} \\
&+ \frac{1}{2} \boldsymbol{x}^{\{i\}T} \langle \mathrm{diag}\left( v_\epsilon^{\{i\}} + \boldsymbol{v}_{\boldsymbol{\delta}}^{\{i\}} \right)^{-1} \begin{bmatrix} \boldsymbol{D} & \boldsymbol{I} \end{bmatrix} \boldsymbol{K}^{\{i\}-1} \begin{bmatrix} \boldsymbol{D}^T \\ \boldsymbol{I} \end{bmatrix} \mathrm{diag}\left( v_\epsilon^{\{i\}} + \boldsymbol{v}_{\boldsymbol{\delta}}^{\{i\}} \right)^{-1} \rangle \boldsymbol{x}^{\{i\}},
\end{aligned}
\tag{53}
$$

where $\boldsymbol{K}^{\{i\}}$ is defined in (29), and $\langle \cdot \rangle$ denotes the expectation over the $q$ distributions. Note that the term $\langle \log \det(\beta_0 \boldsymbol{I} + \beta_1 \boldsymbol{K}_{tp}) \rangle$ can be further simplified as:

$$
\langle \log \det(\beta_0 \boldsymbol{I} + \beta_1 \boldsymbol{K}_{tp}) \rangle = \sum_{t=1}^{L} \langle \log(\beta_0 + \beta_1 r_t) \rangle,
\tag{54}
$$

where $r_t$ denotes the $t$-th eigenvalue of $\boldsymbol{K}_{tp}$. We therefore use this simplification when computing the above expectation. In addition, the expectations in (53) do not have close-form expressions, and can only be computed via Monte-Carlo approximations. More specifically for those variables whose $q$ distributions are log-normal, we can use the reparameterization trick introduced in (Kingma & Welling, 2014). Take $v_\epsilon^{\{i\}} \sim \mathrm{Lognormal}(m_\epsilon^{\{i\}}(\boldsymbol{x}^{\{i\}}), \nu_\epsilon^{\{i\}}(\boldsymbol{x}^{\{i\}}))$ as an example,

$$
\hat{v}_\epsilon^{\{i\}} = \sqrt{\nu_\epsilon^{\{i\}}(\boldsymbol{x}^{\{i\}})} e + m_\epsilon^{\{i\}}(\boldsymbol{x}^{\{i\}}),
\tag{55}
$$

where $e \sim \mathcal{N}(0, 1)$. On the other hand, for $\omega_j \sim \mathrm{Kumaraswamy}(\omega_j; c_{\omega j}(\boldsymbol{x}^{\{1:N\}}), d_{\omega j}(\boldsymbol{x}^{\{1:N\}}))$, samples of $\omega_j$ can be reparameterized as:

$$
\hat{\omega}_j = \left( 1 - u^{\frac{1}{d_{\omega j}(\boldsymbol{x}^{\{1:N\}})}} \right)^{\frac{1}{c_{\omega j}(\boldsymbol{x}^{\{1:N\}})}},
\tag{56}
$$

where $u \sim U(0, 1)$.

The second and third terms $D_{\mathrm{KL}}(q(v_\epsilon^{\{i\}} | \boldsymbol{x}^{\{i\}}) \| p(v_\epsilon^{\{i\}}))$ and $D_{\mathrm{KL}}(q(\boldsymbol{v}_{\boldsymbol{\delta}}^{\{i\}} | \boldsymbol{x}^{\{i\}}) \| p(\boldsymbol{v}_{\boldsymbol{\delta}}^{\{i\}}))$ are the KL divergence between a log-normal distribution and an inverse Gamma distribution. Suppose that the mean and variance of the log-normal distribution are $m$ and $\nu$ and the shape and scale of the inverse gamma distribution are $a$ and $b$. The resulting KL divergence (after removing the constant terms) can be expressed as:

$$
am + b \exp\left( -m + \frac{1}{2}\nu \right) - \frac{1}{2} \log \nu.
\tag{57}
$$

The fourth term $D_{\mathrm{KL}}(q(\omega_j | \boldsymbol{x}^{\{1:N\}}) \| p(\omega_j))$ is the KL divergence between a Kumaraswamy (Nalisnick & Smyth, 2017) with shape parameters $c$ and $d$ and a Beta distribution with shape parameters $a$

**Algorithm 1** AmortizedPeriod

---

**Require:** $N$ time series $\boldsymbol{x}^{\{1:N\}}$ that share common periods, each with length $L$, and the Ramanujan dictionary $\boldsymbol{D}$ with size $\Phi$;
**Ensure:** $q(\omega_j|\boldsymbol{x})$ for $j = 1, \cdots, \Phi$;
 1: **repeat**
 2:      Pass $\boldsymbol{x}^{\{1:N\}}$ and $\boldsymbol{D}$ through the inference network to get the parameters for $q(\omega_j|\boldsymbol{x}^{\{1:N\}})$, $q(\lambda|\boldsymbol{x}^{\{1:N\}})$, $q(\beta_0|\boldsymbol{x}^{\{1:N\}})$, $q(\beta_1|\boldsymbol{x}^{\{1:N\}})$, $q(v_\epsilon^{\{i\}}|\boldsymbol{x}^{\{i\}})$, and $q(\boldsymbol{v}_{\boldsymbol{\delta}}^{\{i\}}|\boldsymbol{x}^{\{i\}})$;
 3:      Draw $\ell$ samples from the Kumaraswamy distribution $q(\omega_j|\boldsymbol{x}^{\{1:N\}})$ using the reparameterization trick in (56);
 4:      Draw $\ell$ samples from the log-normal distributions $q(\lambda|\boldsymbol{x}^{\{1:N\}})$, $q(\beta_0|\boldsymbol{x}^{\{1:N\}})$, $q(\beta_1|\boldsymbol{x}^{\{1:N\}})$, $q(v_\epsilon^{\{i\}}|\boldsymbol{x}^{\{i\}})$, and $q(\boldsymbol{v}_{\boldsymbol{\delta}}^{\{i\}}|\boldsymbol{x}^{\{i\}})$ using the reparameterization trick in (55);
 5:      Draw $\ell$ samples from $q(\boldsymbol{\alpha}^{\{i\}}, \boldsymbol{\tau}^{\{i\}}|\boldsymbol{x}^{\{i\}}, \boldsymbol{\omega}, \lambda, \beta_0, \beta_1, v_\epsilon^{\{i\}}, \boldsymbol{v}_{\boldsymbol{\delta}}^{\{i\}})$ given the samples in the above two steps using the reparameterization trick for Gaussian distributions (Kingma & Welling, 2014);
 6:      Compute the negative ELBO via Monte Carlo approximations (52);
 7:      Update the parameters of the inference network via gradient descent;
 8: **until** convergence;
 9: Perform inference by computing $q(\omega_j|\boldsymbol{x})$ via the inference network for new time series;

---

and $b$, that is,

$$\mathbb{E}_q\Big[\log q(\omega_j|\boldsymbol{x}^{\{1:N\}}) - \log p(\omega_j)\Big]$$

$$= \log c + \log d + (d-1)\langle\log(1-\omega_j^c)\rangle - (a-c)\langle\log\omega_j\rangle - (b-1)\langle\log(1-\omega_j)\rangle, \tag{58}$$

$$= \log c + \log d + \frac{1}{d} - \Big(1 - \frac{a}{c}\Big)\Big(\psi(d) + \frac{1}{d} + \gamma\Big) - (b-1)\langle\log(1-\omega_j)\rangle, \tag{59}$$

where $\psi$ denotes the digamma function, $\gamma$ denotes the Euler–Mascheroni constant, and (59) holds since

$$\langle\log\omega_j\rangle = -\frac{1}{c}\Big(\psi(d) + \frac{1}{d} + \gamma\Big), \tag{60}$$

$$\langle\log(1-\omega_j^c)\rangle = -\frac{1}{d}. \tag{61}$$

Note that the expectation $\langle\log(1-\omega_j)\rangle$ cannot be computed analytically, and so Monte Carlo estimates are required by sampling from the Kumaraswamy distribution as in (56).

The last three terms $D_{\text{KL}}(q(\lambda|\boldsymbol{x}^{\{1:N\}})\|p(\lambda))$, $D_{\text{KL}}(q(\beta_0|\boldsymbol{x}^{\{1:N\}})\|p(\beta_0))$, and $D_{\text{KL}}(q(\beta_1|\boldsymbol{x}^{\{1:N\}})\|p(\beta_1))$ are the KL divergence between the log-normal distribution with mean $m$ and variance $\nu$ and the Gamma distribution with shape $a$ and rate $b$:

$$-am + b\exp\Big(m + \frac{\nu}{2}\Big) - \frac{1}{2}\log\nu. \tag{62}$$

The training and inference algorithm for AmortizedPeriod is summarized in Algorithm 1.

## G   Dataset Description

Here, we provide the details regarding the datasets used in our experiments.

- **Synthetic Data**: We generate periodic synthetic data with noise, outliers, and changing trends. As a first step, we determine the number of periodic components in a time series by randomly picking an integer from $\{0, \cdots, 3\}$, where 0 means there are no periodic components and so $\omega_j$ should be 0 for all $j$. For each periodic components, we further choose the shape of the base periodic signal from sinusoidal, square, and triangle waves at random, all of which are frequently seen in the real world. The corresponding periods are chosen from $\{1, \cdots, 28\}$. We further multiply the chosen periods with 24, corresponding to 24 hours in a day. The amplitude of the base signal is sampled uniformly from $[1, 1.5]$. We also add an irregular monthly period to 25% of the time series. On the other hand, the trend component is also randomly selected as a sinusoidal, square, or triangle wave, with a period as long as the length of the time series $L$. We specify

$L = 120 \text{ days} \times 24 \text{ hours} = 2880$ such that the monthly period component can repeat for about four times. The corresponding amplitude is sampled uniformly from $[5, 10]$. Finally, we set the variance of the Gaussian white noise $v_\epsilon$ and the proportion of outliers $\delta\%$ to be $(0.1, 0.05)$ and $(1, 0.1)$ respectively to generate two datasets, in order to evaluate the periodicity detection approaches under mild and severe conditions. For each dataset, we simulate 5000 samples and split them into training and testing sets with a ratio of $4 : 1$. In particular for AmortizedPeriod, each sample is further partitioned into $N = 24$ time series with the same periods, as mentioned in Section 3.2. We depict a random selection of four time series for both datasets and their corresponding decomposition in Figure 5 and 6. We observe that the periodic components are heavily corrupted by trends, noise, and outliers, especially for the dataset with $v_\epsilon = 1$ and $\delta\% = 0.1$.

- **Yahoo Data**[2]: We utilize the publicly available multiple-period data from Yahoo's webscope S5 datasets, specifically the Yahoo-A3 and Yahoo-A4 datasets. These datasets comprise 200 time series in total, each of which contains 1680 data points across 3 period lengths: 12, 24, and 168. The ratio between the training and testing set is $4 : 1$. Similar to the synthetic data, we divide each sample into $N = 12$ time series with the same periods for AmortizedPeriod. We plot out in Figure 7 four randomly selected from this dataset. It can seen that the noise level associated with time series in this dataset is the lowest in comparison with the other three datasets.

- **App Flow Data**[34]: We further collect the real traffic flow data for 311 micro-services deployed on 18 logic data centers in the cloud system of Ant Group. These data are used for workload forecasting, anomaly detection, and auto-scaling of cloud databases and computing in practice. This dataset consists of 1000 time series, each of length 2880 ($120 \text{ days} \times 24 \text{ hours}$). The candidate periods include 0 (i.e., non-periodic), 1, 7, and 15 days as well as the irregular monthly periods. It can be observed that the periods are corrupted by changes in the periodic and trend patterns, and the presence of noise, outliers, and even blocks of missing data. All time series are labeled manually by experts. The ratio between the training and testing set is $4 : 1$, and we again split each sample into $N = 24$ time series with the same periods for AmortizedPeriod. We further present four time series from this dataset selected at random in Figure 8. As demonstrated in this figure, the trend of the time series is relatively stable. However, there exist outliers every now and then. Moreover, we can observe weekly (the second figure) and monthly periods (the third figure) in this dataset. It is important to note that the monthly pattern observed in the traffic flow of Alipay users can be attributed to activities such as pre-authorized debit, auto-deduction of utility bills (e.g., electricity, water), and auto-payment of credit card bills. Although these activities occur sporadically, they may affect all Alipay users within a brief time frame. As a result, it is essential to consider monthly patterns during resource scheduling, since they may significantly increase the traffic flow.

## H   EXPERIMENT SETUP

Unless otherwise specified, in all of our experiments, we set the hidden dimension in the CAB and SAB to 192. For optimization, we use Adabelief (Zhuang et al., 2020) with $\beta_1 = 0.5$, $\beta_2 = 0.999$, and learning rate $1 \times 10^{-4}$, since it offers more stable results than Adam. We conduct training for 5000 epochs and select the checkpoints with the lowest training loss as the final model. All simulations of training AmortizedPeriod are run using 8 NVIDIA TESLA A100 GPUs with 80 GB of VRAM. All inferences are run on a MacBook Pro (16-inch, 2019) with a 6-core Intel i7 CPU and 16 GB of RAM. All experiments are averaged over 5 trials.

---

[2]https://webscope.sandbox.yahoo.com/catalog.php?datatype=sguccounter=1

[3]https://github.com/alipay/AmortizedPeriod/data/AntData/appflow_data.zip.

[4]a. The data set does not contain any Personal Identifiable Information (PII); b. The data set is desensitized and encrypted; c. Adequate data protection was carried out during the experiment to prevent the risk of data copy leakage, and the data set was destroyed after the experiment; d. The data set is only used for academic research, it does not represent any real business situation.

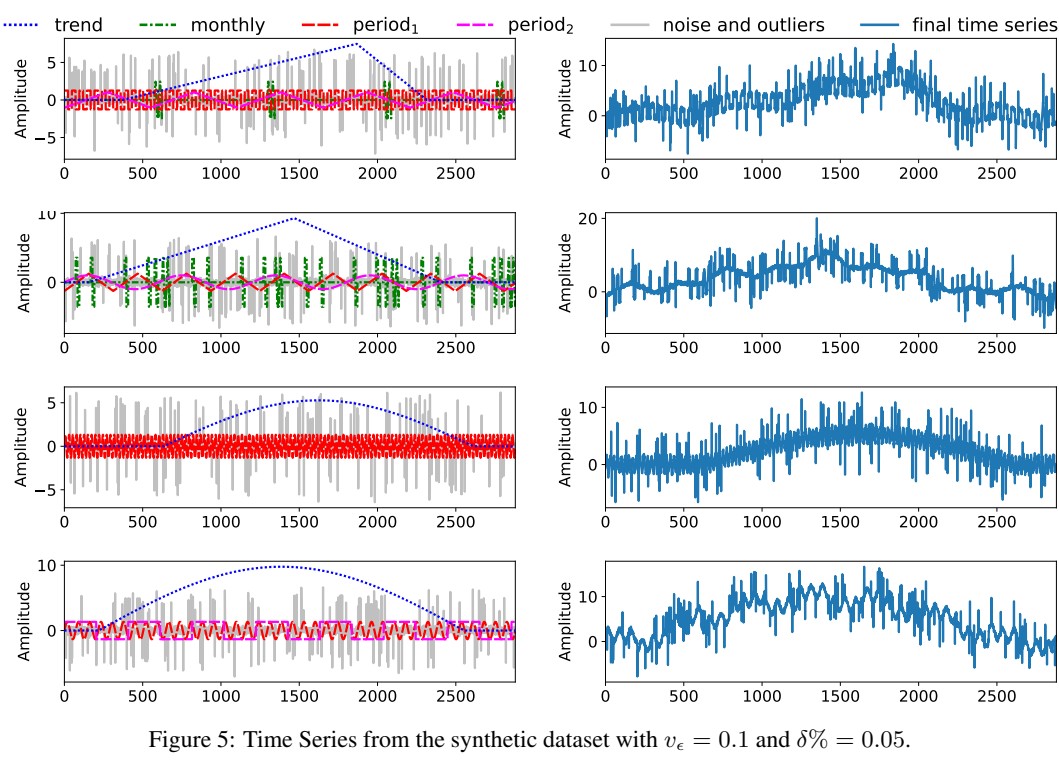

Figure 5: Time Series from the synthetic dataset with $v_\epsilon = 0.1$ and $\delta\% = 0.05$.

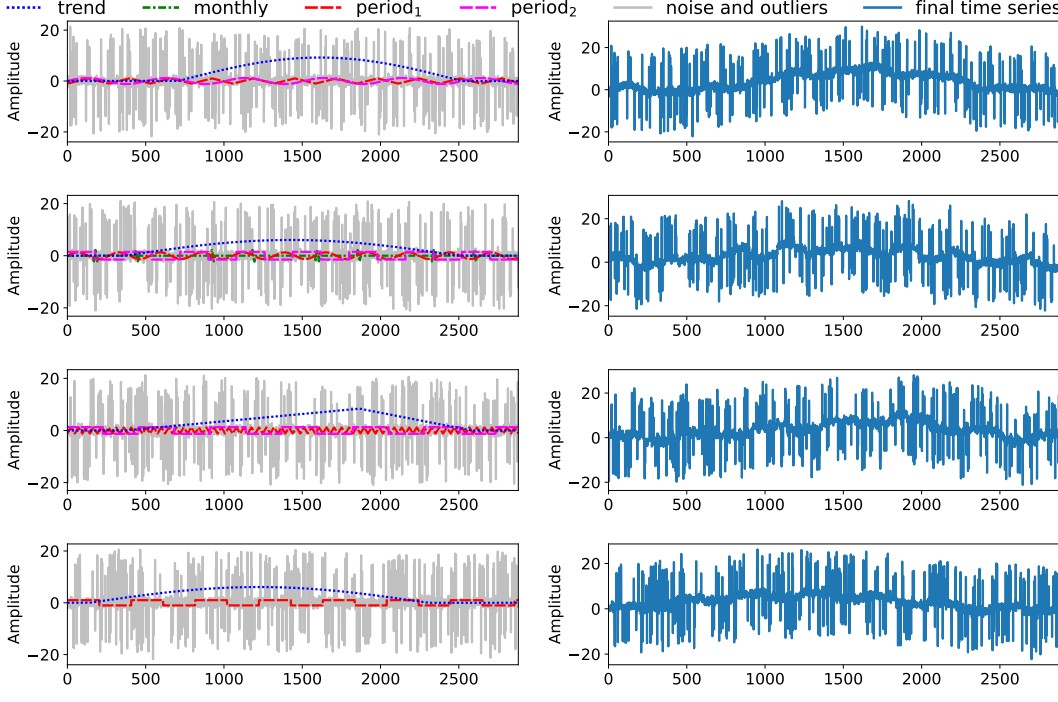

Figure 6: Time Series from the synthetic dataset with $v_\epsilon = 1$ and $\delta\% = 0.1$.

# I EXPERIMENTAL RESULTS

In this section, we present a more comprehensive comparison between AutoPeriod, RobustPeriod, RPT, and the proposed AmortizedPeriod over the four datasets, as a supplement to the results presented in Table 1. Concretely, apart from the micro $F_1$-score reported in Table 1, we also employ other performance metrics to evaluate the benchmark methods, namely the macro $F_1$-score, weighted

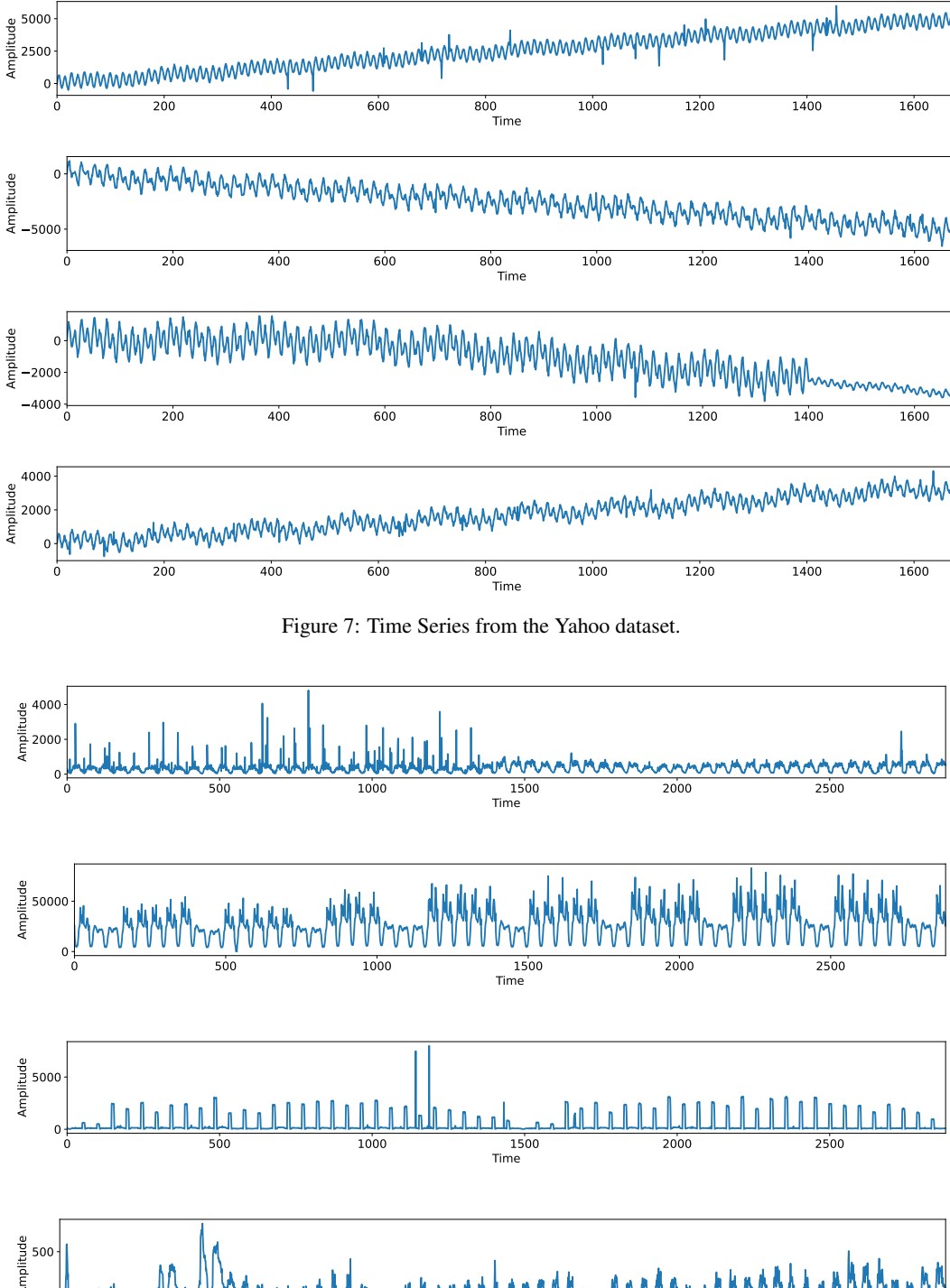

Figure 7: Time Series from the Yahoo dataset.

Figure 8: Time Series with a span of two months from the App Flow dataset.

$F_1$-score, and sampled $F_1$-score. Prior to presenting our findings, we provide an explanation of these scores. The $F_1$-score is a widely used metric for assessing classification model performance, combining precision and recall into a single value that represents their balance. The different variations of the $F_1$-score are defined as follows:

Table 5: Full comparison between the benchmark methods on the four datasets.

| | Metrics | | AutoPeriod | RobustPeriod | RPD | AmortizedPeriod |
|---|---|---|---|---|---|---|
| $v_e = 0.1, \delta\% = 0.05$ | Micro $F_1$-score | train | 0.29 | 0.32 | 0.24 | **0.80** |
| | | test | 0.30 | 0.30 | 0.23 | **0.76** |
| | Macro $F_1$-score | train | 0.18 | 0.19 | 0.18 | **0.65** |
| | | test | 0.17 | 0.19 | 0.18 | **0.57** |
| | Weighted $F_1$-score | train | 0.30 | 0.31 | 0.39 | **0.87** |
| | | test | 0.29 | 0.29 | 0.39 | **0.84** |
| | Sampled $F_1$-score | train | 0.23 | 0.29 | 0.23 | **0.80** |
| | | test | 0.22 | 0.27 | 0.22 | **0.75** |
| | Time | | 1.76e-1 | 5.91e2 | 5.39 | **3.23e-2** |
| | Memory | | 2.57MB | 609.48MB | 144.60MB | **2.79KB** |
| $v_e = 1, \delta\% = 0.1$ | Micro $F_1$-score | train | 0.29 | 0.31 | 0.18 | **0.79** |
| | | test | 0.30 | 0.29 | 0.17 | **0.57** |
| | Macro $F_1$-score | train | 0.18 | 0.18 | 0.10 | **0.58** |
| | | test | 0.18 | 0.17 | 0.10 | **0.45** |
| | Weighted $F_1$-score | train | 0.27 | 0.28 | 0.22 | **0.86** |
| | | test | 0.27 | 0.26 | 0.22 | **0.72** |
| | Sampled $F_1$-score | train | 0.22 | 0.28 | 0.12 | **0.73** |
| | | test | 0.22 | 0.26 | 0.12 | **0.54** |
| | Time | | 9.88e-2 | 4.41e2 | 1.24e2 | **3.72e2** |
| | Memory | | 2.56MB | 93.98MB | 144.60MB | **2.90KB** |
| Yahoo | Micro $F_1$-score | train | 0.75 | 0.84 | 0.83 | **0.86** |
| | | test | 0.73 | 0.82 | 0.84 | **0.87** |
| | Macro $F_1$-score | train | 0.29 | 0.09 | 0.59 | **0.81** |
| | | test | 0.23 | 0.12 | 0.38 | **0.82** |
| | Weighted $F_1$-score | train | 0.87 | **0.98** | 0.81 | 0.81 |
| | | test | 0.92 | **0.98** | 0.84 | 0.82 |
| | Sampled $F_1$-score | train | 0.69 | 0.84 | 0.82 | **0.86** |
| | | test | 0.71 | 0.83 | 0.84 | **0.86** |
| | Time | | 1.50e-1 | 1.24e2 | 1.17 | **3.48e-2** |
| | Memory | | 1.33MB | 50.20MB | 35.87MB | **2.31KB** |
| App Flow | Micro $F_1$-score | train | 0.40 | 0.57 | 0.59 | **0.88** |
| | | test | 0.45 | 0.56 | 0.60 | **0.85** |
| | Macro $F_1$-score | train | 0.03 | 0.07 | 0.04 | **0.28** |
| | | test | 0.04 | 0.08 | 0.06 | **0.26** |
| | Weighted $F_1$-score | train | 0.39 | 0.78 | 0.62 | **0.90** |
| | | test | 0.46 | 0.82 | 0.65 | **0.88** |
| | Sampled $F_1$-score | train | 0.30 | 0.55 | 0.62 | **0.82** |
| | | test | 0.36 | 0.55 | 0.63 | **0.79** |
| | Time | | 7.68e-2 | 3.72e2 | 8.47 | **3.43e-2** |
| | Memory | | 2.57MB | 94.17MB | 144.61MB | **2.66KB** |

- Micro $F_1$-score: This metric considers the overall count of true positives, false positives, and false negatives across all classes. It computes the precision and recall for the entire dataset and then derives the $F_1$-score based on these aggregated values. It treats all classes equally and is suitable for imbalanced datasets.

- Macro $F_1$-score: This metric calculates the $F_1$-score for each class individually and then takes the average across all classes. Each class is given equal importance, irrespective of its size or distribution in the dataset. It is appropriate when all classes are considered equally significant.

- Weighted $F_1$-score: Similar to the macro $F_1$-score, this metric accounts for the class distribution within the dataset. It calculates the $F_1$-score for each class, weighting them by the number of

Table 6: The micro $F_1$-score and run time (seconds) averaged over all time series resulting from all benchmark methods for four datasets. The first two datasets consist of synthetic data with different noise variance $v_\epsilon$ and proportion of outliers $\delta\%$. The results are averaged over 5 trials and the corresponding standard deviation is provided in the brackets below.

| Methods | $v_\epsilon = 0.1, \delta\% = 0.05$ | | | $v_\epsilon = 1, \delta\% = 0.1$ | | | Yahoo | | | App Flow | | |
|---|---|---|---|---|---|---|---|---|---|---|---|---|
| | Train | Test | Time | Train | Test | Time | Train | Test | Time | Train | Test | Time |
| AutoPeriod | 0.29 | 0.30 | 1.76e-1 | 0.29 | 0.28 | 9.88e-2 | 0.75 | 0.73 | 1.50e-1 | 0.40 | 0.45 | 7.68e-2 |
| | (0) | (0) | (1.67e-2) | (0) | (0) | (2.11e-2) | (0) | (0) | (7.17e-3) | (0) | (0) | (4.98e-3) |
| RobustPeriod | 0.32 | 0.30 | 5.91e2 | 0.31 | 0.29 | 4.41e2 | 0.84 | 0.82 | 1.24e2 | 0.57 | 0.56 | 3.72e2 |
| | (1.54e-3) | (7.75e-4) | (3.76e1) | (3.88e-3) | (2.31e-3) | (4.41e1) | (0) | (0) | (7.33) | (0) | (0) | (5.06e1) |
| RPD | 0.24 | 0.23 | 5.39 | 0.18 | 0.17 | 5.06 | 0.83 | 0.84 | 1.17 | 0.59 | 0.60 | 8.47 |
| | (5.98e-5) | (7.59e-5) | (4.59e-1) | (1.89e-5) | (7.42e-5) | (4.31e-1) | (0) | (0) | (9.52e-2) | (4.46e-5) | (0) | (2.68e-1) |
| AmortizedPeriod | 0.80 | 0.76 | 3.23e-2 | 0.70 | 0.57 | 3.32e-2 | 0.86 | 0.87 | 3.48e-2 | 0.90 | 0.85 | 3.43e-2 |
| | (1.62e-2) | (2.02e-2) | (2.10e-3) | (1.58e-2) | (9.37e-3) | (4.69e-3) | (2.36e-2) | (2.54e-2) | (2.92e-3) | (2.00e-2) | (2.57e-2) | (4.11e-3) |

instances in each class, and then takes the average. This evaluation provides a more balanced assessment when the classes have varying sizes.

- Sampled $F_1$-score: Comparable to the macro $F_1$-score, this metric calculates the $F_1$-score for each individual sample and subsequently takes the average across all samples. It assigns equal importance to each sample, irrespective of its class or distribution. This metric is useful for evaluating classifier performance on a per-sample basis, without considering class imbalances or class-specific performance.

Moreover, in addition to the running time shown in Table 1, we also show the peak memory consumption during inference. All of these results are summarized in Table 5.

In terms of estimation accuracy, AmortizedPeriod demonstrates superior performance compared to other methods for most datasets and variants of the $F_1$-score, establishing its dominance. However, for the Yahoo dataset, RobustPeriod surpasses AmortizedPeriod in terms of the weighted $F_1$-score while falling short in other $F_1$-score variants. This discrepancy can be attributed to RobustPeriod generating more false positives for nonexistent periods, yet yielding fewer false negatives for existing periods. Notably, when computing the weighted $F_1$-score, the weights assigned to non-existing periods are set to zero, thereby disregarding the false positives produced by RobustPeriod and inflating the resulting weighted $F_1$-score.

In terms of efficiency, AmortizedPeriod exhibits remarkable superiority over other methods, as it requires the least running time and memory resources. Unlike RobustPeriod and RPT, which involve iterative optimization algorithms with computationally intensive operations such as matrix inverses for accurate period estimation, AmortizedPeriod solely relies on the inference network that takes the original time series as input and outputs the estimated periods. Specifically, we focus exclusively on inferring the existence of a period (i.e., $q(\boldsymbol{\omega})$) during inference, making the distribution of other components irrelevant. Thus, we employ only the inference network within the red dashed box depicted in Figure 2, resulting in a lightweight network architecture. Additionally, when applying AmortizedPeriod, we leverage a strategy where the original time series is divided into $N$ subseries with identical periods, based on the base periods of the time series. This approach is elaborated upon at the end of Section 3.2 and in Appendix G when introducing the datasets. For instance, consider a time series spanning 120 days, with each day comprising $N = 24$ time points. By aggregating the $i$-th time point across consecutive days, we create $N = 24$ time series, each with a length of 120. Consequently, we can collectively detect common periods (e.g., daily, weekly, and monthly) for these 24 time series. This practice effectively reduces the number of potential bases in the Ramanujan dictionary $\boldsymbol{D}$. Recall that the attention is computed between the set of time series and the basis vectors, and so shrinking $\boldsymbol{D}$ can enhance efficiency. Unfortunately, this strategy is not applicable to other methods, as they can only detect periods for a single time series instead of identifying common periods among a set of time series.

Furthermore, we report the standard deviation computed over five trials in Table 6 and 7. Our analysis reveals that the standard deviation for all methods is relatively small in comparison to the mean, especially for the SOTA methods.

Table 7: The micro $F_1$-score for the ablation study. The results are averaged over 5 trials and the corresponding standard deviation is provided in the brackets below.

| Methods | $v_\epsilon = 0.1, \delta\% = 0.05$ | | $v_\epsilon = 1, \delta\% = 0.1$ | | Yahoo | | App Flow | |
|---|---|---|---|---|---|---|---|---|
| | Train | Test | Train | Test | Train | Test | Train | Test |
| AmortizedPeriod | 0.80 | 0.76 | 0.70 | 0.57 | 0.86 | 0.87 | 0.89 | 0.87 |
| | (1.62e-2) | (2.02e-2) | (1.58e-2) | (9.37e-3) | (2.36e-2) | (2.54e-2) | (2.00e-2) | (2.57e-2) |
| - Trend | 0.58 | 0.55 | 0.58 | 0.48 | 0.85 | 0.86 | 0.88 | 0.86 |
| | (7.52e-3) | (7.13e-3) | (4.33e-3) | (3.16e-3) | (2.87e-2) | (6.29e-3) | (5.61e-3) | (1.11e-2) |
| - Outliers | 0.51 | 0.48 | 0.28 | 0.29 | 0.86 | 0.86 | 0.55 | 0.50 |
| | (2.75e-2) | (4.83e-2) | (2.91e-2) | (2.85e-2) | (1.11e-3) | (3.12e-3) | (4.88e-2) | (5.12e-2) |
| - Month | 0.71 | 0.70 | 0.53 | 0.49 | - | - | 0.85 | 0.84 |
| | (1.88e-2) | (2.62e-2) | (1.62e-2) | (1.21e-2) | - | - | (3.84e-2) | (5.51e-2) |

Table 8: Full comparison between the benchmark methods on the CRAN dataset.

| Metrics | | AutoPeriod | RobustPeriod | RPD | AmortizedPeriod |
|---|---|---|---|---|---|
| Micro $F_1$-score | train | 0.05 | 0.23 | 0.26 | **0.50** |
| | test | 0.12 | 0.18 | 0.20 | **0.25** |
| Macro $F_1$-score | train | 0.02 | 0.13 | 0.12 | **0.25** |
| | test | 0.08 | **0.12** | 0.10 | **0.12** |
| Weighted $F_1$-score | train | 0.05 | 0.42 | 0.49 | **0.51** |
| | test | 0.06 | 0.31 | **0.46** | 0.18 |
| Sampled $F_1$-score | train | 0.02 | 0.25 | 0.34 | **0.50** |
| | test | 0.06 | 0.17 | **0.26** | 0.25 |
| Time | | 1.76e-1 | 4.96e2 | 1.14e1 | **1.47** |
| Memory | | 2.57MB | 29.84MB | 103.36MB | **2.40KB** |

## J  ADDITIONAL EXPERIMENTS ON THE CRAN DATASET WITH SINGLE PERIODS

In this section, we evaluate the four methods for period detection using the CRAN data. The dataset consists of univariate time series extracted from open-source packages listed in the "Time Series Data" section of the CRAN Task View on Time Series Analysis Toller et al. (2019) [5]. It includes 82 time series in a diverse range of application domains, including economic indicators like employment rates and retail sales, as well as environmental measurements such as pollution levels and sunspot counts.

It should be noted that the time series within this dataset have varying lengths. To enable the application of AmortizedPeriod to this dataset, a preprocessing step is undertaken involving linear interpolation and truncation. This ensures that all resulting time series have the same length of 1040. Specifically, if a time series $x^{\{i\}}$ has a length $L^{\{i\}}$ less than 1040, we perform linear interpolation by introducing $\lfloor 1040/L^{\{i\}} \rfloor + 1$ new points between each pair of consecutive time points in the original series, where $\lfloor \cdot \rfloor$ represents the floor function. When $L^{\{i\}}$ is greater than or equal to 1040, we truncate the original series, retaining the first 1040 time points. Following this preprocessing step, it is observed that the true periods for all time series are multiples of 8, with the exception of two series[6]. Consequently, these two series are excluded from the analysis. The resulting periods for all remaining time series range from 8 to 528. Similar to the aforementioned datasets, each sample is partitioned into $N = 8$ time series, each sharing the same period. The training and testing sets maintain a ratio of 4:1, respectively. For AmortizedPeriod, we train the inference network with 5e4 epochs and choose the checkpoints with the lowest training loss. The remaining settings are the same as before.

---

[5]https://cran.r-project.org/web/views/TimeSeries.html
[6]These two series initially have periods of 7 and 52.1786, respectively.

The results of our experiments are presented in Table 8, where it can be observed that the proposed AmortizedPeriod consistently outperforms other methods in terms of accuracy and efficiency across most evaluation criteria. It is worth noting that the performance of AmortizedPeriod on the testing data is comparatively lower than that on the training data. This discrepancy arises due to the limited number of time series in the training set (i.e., 64), which restricts the generalization capability of the learned inference network to unseen testing data. Nevertheless, we would like to emphasize that the periods for the training data are estimated in a self-supervised manner. Consequently, the superior $F_1$-score achieved by AmortizedPeriod indicates the versatility of the proposed generative model in describing the training data, while the inference model effectively maps the observed time series to the distribution of the periods.

On the other hand, the hyperparameters for AutoPeriod and RobustPeriod were set to be the same as those used in the four datasets discussed in Section 4. These parameters were manually tuned to yield good performance on those datasets. Although tuning these parameters specifically for the CRAN data might potentially enhance their performance, it would require additional effort. This underscores the importance of adaptive hyperparameter learning from the data, a feature effectively addressed by AmortizedPeriod.

Finally, for RPD, we employed cross-validation to select the single hyperparameter in all our experiments, which contributes to its relatively robust performance. However, since RPD does not account for the non-periodic components in the data, its performance remains inferior to that of AmortizedPeriod, particularly for the training data.

## K  LIMITATIONS, DISCUSSIONS, AND FUTURE WORK

### K.1  LIMITATIONS OF AMORTIZEDPERIOD

First, AmortizedPeriod is more suitable for scenarios where there are hundreds or thousands of time series, as some training data is required to train the inference network. When there are only a few time series, we recommend utilizing the existing SOTA methods. In addition, before using it for inference, AmortizedPeriod needs to be trained for a relatively long time. When the training dataset is larger, the training process can be time-consuming. However, after AmortizedPeriod is well-trained, the inference process is relatively fast, as shown in our experiments.

### K.2  DISCUSSIONS ON THE USAGE OF PERIODICITY DETECTION

As mentioned in Section 1, periodicity detection offers benefits for other time series-related tasks, such as classification, clustering, decomposition, anomaly detection, and forecasting. In this regard, we will now delve deeper into how accurately estimated periods can boost the performance of these tasks.

- Time series classification and clustering: Periodic patterns in time series data often contain informative and discriminative features that can distinguish different classes or clusters with the noise and irrelevant variations in time series filter out. For example, in heart rate monitoring, periodicity detection can identify recurring patterns, such as beat-to-beat intervals, enabling the differentiation between normal heart rhythms and irregularities related to cardiac conditions (Thungtong et al., 2018). Similarly, in customer behavior analysis, periodicity detection helps identify regular patterns in purchasing behavior, facilitating the clustering of customers based on their shopping habits (Guidotti et al., 2018).

- Time series decomposition: Many typical time series decomposition methods, such as MSTL (Bandara et al., 2021) and robustSTL (Wen et al., 2019), rely on predefined periods for extracting seasonal components. By accurately detecting periodic patterns, we can extract underlying seasonal variations more effectively, leading to a cleaner and more accurate decomposition.

- Anomaly detection: Periodicity detection aids in accurately identifying and characterizing normal patterns or behaviors in time series data. Establishing a baseline for normal behavior based on the underlying periodic patterns enables the detection of anomalies or deviations from expected periodic behavior (Shehu & Harper, 2023). For instance, in network traffic analysis, periodicity detection helps identify regular communication and data transfer patterns, enabling the identification of anomalies that deviate from the expected periodic behavior (Akpinar & Ozcelik, 2020).

- Time series forecasting: Periodicity detection allows for the identification and modeling of seasonal, trend, and noise patterns in time series data, as mentioned above. By accurately capturing the underlying periodic components, we can extract and model seasonal and trend variations more accurately, while reducing the influence of unpredictable noise. Periodicity patterns have been exploited explicitly or implicitly in various forecasting algorithms, including ARIMA (Box & Jenkins, 1968), prophet (Taylor & Letham, 2018), N-BEATS (Oreshkin et al., 2019), Pyraformer (Liu et al., 2022), TimesNet (Wu et al., 2023), etc.

### K.3 FUTURE WORK

In future research, an intriguing avenue to explore is the concept of pre-training in the context of time series analysis. Specifically, we can investigate the feasibility of pre-training an inference network using a large and diverse set of time series data. This pre-trained network can then be directly applied to perform periodicity detection without requiring additional training. Additionally, it would be valuable to assess whether the features extracted by the pre-trained inference network are also effective for other time series-related tasks, such as classification and forecasting. To this end, the pre-trained inference network could be utilized for downstream tasks with some fine-tuning, further investigating its potential for enhancing performance across various time series analysis applications.

