# OpenReview forum: "AmortizedPeriod: Attention-based Amortized Inference for Periodicity Identification"
_ICLR.cc/2024/Conference — ICLR 2024 poster_

### Official Review · Reviewer_KTuk · 2023-10-27

**Soundness:** 3 good
**Presentation:** 3 good
**Contribution:** 3 good
**Rating:** 6
**Confidence:** 4

**Summary:**

This paper focuses on the periodicity detection problem, and proposed a model AmortizedPeriod which can address the limited robustness and memorylessness problems in SOTA by applying Bayesian paradigm and variational inference. Extensive experiments are conducted.

**Strengths:**

1. The paper is well organized and well presented.
2. A novel model AmortizedPeriod is proposed which can address the limited robustness and memorylessness problems in the existing works.
3. Extensive experiments are conducted to validate the model effectiveness, the results look promising.

**Weaknesses:**

1. Since the proposed model is trying to deal with time series data with self-attention, which also requires a lot of memory, there should be some experiments to compare the memories used by the proposed method and other methods.
2. In the experiment, the results of multiple periodicity scenarios should be provided, for example, when the data has multiple periodicity, irregular periods, noise, outliers, etc.

**Questions:**

Please address the questions above.

---

> ### Author Response · Authors · 2023-11-23
> **Reply to Reviewer KTuk**
>
> We value your constructive and thoughtful comments. We believe that addressing the reviewer’s comments has resulted in improving the clarity and presentation of the paper’s contributions and has brought the paper to a higher standard. We provide a detailed response to each of the reviewer's comments. The reviewer's comments are presented in italics, followed by our response. Quotations from the revised paper are included in markdown quotation mode. The corresponding modifications in the paper are highlighted in blue. Unless otherwise specified, all references to pages, equations, sections, and bibliographic citations relate to the revised paper.
>
> _Q1 - comparison of memory consumption_
>
> Thanks a lot for pointing this out! **We have conducted experiments to compare the memory consumption of the proposed method and the other methods utilized in this study. The results of these experiments are presented and thoroughly discussed in Appendix I, specifically on Pages 26-27 of the paper**, as follows:
>
> > "In addition to the running time shown in Table 1, we also show the peak memory consumption during inference. All of these results are summarized in the table below."
> >
> >| Datasets | AutoPeriod | RobustPeriod | RPT | AmortizedPeriod |
> >| --- | --- | --- | --- | --- |
> >| SynData1 | 2.57MB | 609.48MB | 144.60MB | 2.79KB |
> >| SynData2 | 2.56MB | 93.98MB | 144.60MB | 2.90KB |
> >| Yahoo | 1.33MB | 50.20MB | 35.87MB | 2.31KB |
> >| AppFlow | 2.57MB | 94.17MB | 144.61MB | 2.66KB |
> >
> > "In terms of efficiency, AmortizedPeriod exhibits remarkable superiority over other methods, as it requires the least running time and memory resources. Unlike RobustPeriod and RPT, which involve iterative optimization algorithms with computationally intensive operations such as matrix inverses for accurate period estimation, AmortizedPeriod solely relies on the inference network that takes the original time series as input and outputs the estimated periods. Specifically, we focus exclusively on inferring the existence of a period (i.e., $q(\omega)$) during inference, making the distribution of other components irrelevant. Thus, we employ only the inference network within the red dashed box depicted in Figure 2, resulting in a lightweight network architecture. Additionally, when applying AmortizedPeriod, we leverage a strategy where the original time series is divided into $N$ subseries with identical periods, based on the base periods of the time series. This approach is elaborated upon at the end of Section 3.2 and in Appendix C when introducing the datasets. For instance, consider a time series spanning 120 days, with each day comprising $N=24$ time points. By aggregating the $i$-th time point across consecutive days, we create $N=24$ time series, each with a length of 120. Consequently, we can collectively detect common periods (e.g., daily, weekly, and monthly) for these 24 time series. This practice effectively reduces the number of potential bases in the Ramanujan dictionary $D$. **Recall that the attention is computed between the set of time series and the basis vectors, and so shrinking $D$ can enhance efficiency.** Unfortunately, this strategy is not applicable to other methods, as they can only detect periods for a single time series instead of identifying common periods among a set of time series."
>
>
>
> _Q2 - the results of multiple periodicity scenarios_
>
> We have conducted experiments for time series with multiple and irregular periods, trends, noise, and outliers. Time series in two synthetic datasets and the App Flow dataset already have these components. Detailed descriptions of these datasets can be found **in Appendix G, which is located on Pages 22-23** of the paper. To illustrate the diversity of the data, we have randomly selected and visualized some example time series from these datasets in Figures 5-8. These figures provide a visual representation of the various components present in the data.

---

### Official Review · Reviewer_ue83 · 2023-10-30

**Soundness:** 3 good
**Presentation:** 3 good
**Contribution:** 3 good
**Rating:** 6
**Confidence:** 4

**Summary:**

To tackle the problems of limited robustness and lacking memory, the authors proposed AmortizedPeriod for Periodic patterns combining Bayesian statistics and deep learning. The proposed method can model the correlation of the periods, trends, noise, and outliers in time series, and take use of missing data and irregular periods at the same time. The authors also conduct several experiments to prove the effectiveness of the proposed framework.

**Strengths:**

S1. The periodicity detection task is essential and interesting.
S2. The authors provide rigorous theoretical analysis for their proposed methods.
S3. The authors have conducted several experiments to indicate the effectiveness of their proposed methods.

**Weaknesses:**

W1. The authors may include more preliminaries to explain the tasks, such as problem definition. This can be beneficial for readers without sufficient background of periodicity detection tasks to understand the paper.
W2. The authors may include more error metrics to measure the performance of periodicity detection tasks like other classification tasks.
W3. The authors may add more downstream tasks to further evaluate the effectiveness of periodicity detection tasks.

**Questions:**

1.	Can the author include more preliminaries?
2.	Can the author include more error metrics?
3.	Can the author add more downstream tasks?

---

> ### Author Response · Authors · 2023-11-23
> **Reply to Reviewer ue83 (Part 1)**
>
> We sincerely appreciate your constructive and thoughtful comments. We believe that by addressing these comments, we have significantly enhanced the clarity, presentation, and overall quality of the paper, raising it to a higher standard. We provide a detailed response to each of the reviewer's comments. The reviewer's comments are presented in italics, followed by our response. Quotations from the revised paper are included in markdown quotation mode. The corresponding modifications in the paper are highlighted in blue. Unless otherwise specified, all references to pages, equations, sections, and bibliographic citations relate to the revised paper.
>
> _Q1 - more preliminaries_
>
> Thanks for your suggestion! In response to your comment, **we have reorganized Section 3.1 to include necessary preliminaries. We now provide definitions for both $q$-periodic and exactly $q$-periodic time series. Additionally, we explain how the Ramanujan subspaces can be employed to effectively identify the exact periods concealed within a time series.** The updated Section 3.1 is as follows:
> > "The definition of Ramanujan subspaces is cumbersome to describe and not illuminating, so we present instead the definition of $q$-periodic time series as a starting point. From there, we proceed to discuss the properties of Ramanujan subspaces relevant to periodicity detection and define exactly $q$-periodic time series. We refer the readers to Appendix C for the formal definition of Ramanujan subspaces.
>
> > Defintion 1. _($q$-periodic) A time series $s(t)$ with time stamp $t$ of length $L$ ($1\leq t \leq L$) is $q$-periodic if $q$ is the smallest integer such that $s(t + q) = s(t)$ for all $t$, and $q$ is called the period of $s(t)$_.
>
> > This definition eliminates the ambiguity that a time series of period $q$ is also of a period that is a multiple of $q$, which plagues the time-domain methods for periodicity detection. However, another problem comes along with this definition, that is, a time series with an exact period of 6, for instance, may be further decomposed into two periodic components with periods 2 and 3 respectively, making it necessary to detect these fundamental exact periods in practical scenarios. To attack this problem, Vaidyanathan (2014a;b) proposed the Ramanujan subspaces, which construct a distinct linear subspace $\mathcal S_q$ for each period $q$. The time series spanned by a subspace $\mathcal S_q$ can only possess a period of $q$ and cannot be decomposed into smaller periods. Moreover, two arbitrary subspaces $\mathcal S_{q_1}$ and $\mathcal S_{q_2}$ are orthogonal to each other if $q_1 \neq q_2$. As a result, we can define the exact period as:
>
> > Definition 2. _(exactly $q$-periodic (Muresan & Parks, 2003)) A time series $s(t)$ is of exact period $q$ if  $s(t)$ is in the Ramanujan subspace $\mathcal S_q$ and the projection of $s(t)$ onto_ $\mathcal S_\bar{q}$ _is zero for all $\bar{q} < q$_.
>
> > Let $R_q$ be the linear basis for $\mathcal S_q$ and $D = [R_1, \cdots, R_{P_\text{max}}]$ the resulting Ramanujan dictionary with a maximum period of $P_\text{max}$. We can ascertain the periods of a time series $s(t)$ by decomposing it as $s = D\alpha$, where $\alpha$ is the coefficient vector. Ideally, $\alpha$ should have non-zero entries only at locations where the periodic components of $s$ reside. Estimating $\alpha$ given $s$ and $D$ can be formulated as a sparse vector recovery problem, such as basis pursuit or lasso (Tenneti & Vaidyanathan, 2016)."
>
>
> Furthermore, **we have relocated the definition of the Ramanujan subspaces from Section 3.1 to Appendix C.** This decision was made to take advantage of the ample space available in the appendix. In Appendix C, we provide a more comprehensive explanation of the definitions of Ramanujan subspaces and their relevant properties pertaining to periodicity detection. This will allow readers to gain a better understanding of the background knowledge. For further details, please refer to Appendix C.

---

> ### Author Response · Authors · 2023-11-23
> **Reply to Reviewer ue83 (Part 2)**
>
> _Q2 - more error metrics_
>
> Apart from the micro $F_1$-score, we have included macro, weighted, and sampled $F_1$-score. AmortizedPeriod consistently outperforms other methods in terms of these metrics. We have presented the results **in Appendix I on Pages 24-27** as follows:
> > "In this section, we present a more comprehensive comparison between AutoPeriod, RobustPeriod, RPT, and the proposed AmortizedPeriod over the four datasets, as a supplement to the results presented in Table 1.
> >
> > Concretely, apart from the micro $F_1$-score reported in Table 1, we also employ other performance metrics to evaluate the benchmark methods, namely the macro $F_1$-score, weighted $F_1$-score, and sampled $F_1$-score. Prior to presenting our findings, we provide an explanation of these scores. The $F_1$-score is a widely used metric for assessing classification model performance, combining precision and recall into a single value that represents their balance. The different variations of the $F_1$-score are as defined follows:
> > - Micro $F_1$-score: This metric considers the overall count of true positives, false positives, and false negatives across all classes. It computes the precision and recall for the entire dataset and then derives the $F_1$-score based on these aggregate values. It treats all classes equally and is suitable for imbalanced datasets.
> > - Macro $F_1$-score: This metric calculates the $F_1$-score for each class individually and then takes the average across all classes. Each class is given equal importance, irrespective of its size or distribution in the dataset. It is appropriate when all classes are considered equally significant.
> > - Weighted $F_1$-score: Similar to the macro $F_1$-score, this metric accounts for the class distribution within the dataset. It calculates the $F_1$-score for each class, weighting them by the number of instances in each class, and then takes the average. This evaluation provides a more balanced assessment when the classes have varying sizes.
> > - Sampled $F_1$-score: Comparable to the macro $F_1$-score, this metric calculates the $F_1$-score for each individual sample and subsequently takes the average across all samples. It assigns equal importance to each sample, irrespective of its class or distribution. This metric is useful for evaluating classifier performance on a per-sample basis, without considering class imbalances or class-specific performance."
>
>
> > "In terms of estimation accuracy, AmortizedPeriod demonstrates superior performance compared to other methods for most datasets and variants of the $F_1$-score, establishing its dominance. However, for the Yahoo dataset, RobustPeriod surpasses AmortizedPeriod in terms of the weighted $F_1$-score while falling short in other $F_1$-score variants. This discrepancy can be attributed to RobustPeriod generating more false positives for nonexistent periods, yet yielding fewer false negatives for existing periods. Notably, when computing the weighted $F_1$-score, the weights assigned to non-existing periods are set to zero, thereby disregarding the false positives produced by RobustPeriod and inflating the resulting weighted $F_1$-score."
> | Datasets | Metrics |  | AutoPeriod | RobustPeriod | RDP | AmortizedPeriod |
> | --- | --- | --- | --- | --- | --- | --- |
> | SynData1 | Micro F1 | Train | 0.29 | 0.32 | 0.24 | **0.80** |
> |  |  | Test | 0.30 | 0.30 | 0.23 | **0.76** |
> |  | Macro F1 | Train | 0.18 | 0.19 | 0.18 | **0.65** |
> |  |  | Test | 0.17 | 0.19 | 0.18 | **0.57** |
> |  | Weighted F1 | Train | 0.30 | 0.31 | 0.39 | **0.87** |
> |  |  | Test | 0.29 | 0.29 | 0.39 | **0.84** |
> |  | Sampled F1 | Train | 0.23 | 0.29 | 0.23 | **0.80** |
> |  |  | Test | 0.22 | 0.27 | 0.22 | **0.75** |
> | SynData2 | Micro F1 | Train | 0.29 | 0.31 | 0.18 | **0.79** |
> |  |  | Test | 0.30 | 0.29 | 0.17 | **0.57** |
> |  | Macro F1 | Train | 0.18 | 0.18 | 0.10 | **0.58** |
> |  |  | Test | 0.18 | 0.17 | 0.10 | **0.45** |
> |  | Weighted F1 | Train | 0.27 | 0.28 | 0.22 | **0.86** |
> |  |  | Test | 0.27 | 0.26 | 0.22 | **0.72** |
> |  | Sampled F1 | Train | 0.22 | 0.28 | 0.12 | **0.73** |
> |  |  | Test | 0.22 | 0.26 | 0.12 | **0.54** |
> | Yahoo | Micro F1 | Train | 0.75 | 0.84 | 0.83 | **0.86** |
> |  |  | Test | 0.73 | 0.82 | 0.84 | **0.87** |
> |  | Macro F1 | Train | 0.29 | 0.09 | 0.59 | **0.81** |
> |  |  | Test | 0.23 | 0.12 | 0.38 | **0.82** |
> |  | Weighted F1 | Train | 0.87 | **0.98** | 0.81 | 0.81 |
> |  |  | Test | 0.92 | **0.98** | 0.84 | 0.82 |
> |  | Sampled F1 | Train | 0.69 | 0.84 | 0.82 | **0.86** |
> |  |  | Test | 0.71 | 0.83 | 0.84 | **0.86** |
> | AppFlow | Micro F1 | Train | 0.40 | 0.57 | 0.59 | **0.88** |
> |  |  | Test | 0.45 | 0.56 | 0.60 | **0.85** |
> |  | Macro F1 | Train | 0.03 | 0.07 | 0.04 | **0.28** |
> |  |  | Test | 0.04 | 0.08 | 0.06 | **0.26** |
> |  | Weighted F1 | Train | 0.39 | 0.78 | 0.62 | **0.90** |
> |  |  | Test | 0.46 | 0.82 | 0.65 | **0.88** |
> |  | Sampled F1 | Train | 0.30 | 0.55 | 0.62 | **0.82** |
> |  |  | Test | 0.36 | 0.55 | 0.63 | **0.79** |

---

> ### Author Response · Authors · 2023-11-23
> **Reply to Reviewer ue83 (Part 3)**
>
> _Q3 - more downstream tasks_
>
> Given the scope of this paper and the time constraints, we have currently chosen not to include additional downstream tasks. However, **we acknowledge the potential value in exploring further downstream tasks to evaluate the effectiveness of periodicity detection.** In lieu of including these tasks in the present work, **we have provided a comprehensive discussion on how accurate periodicity detection can enhance the performance of related downstream tasks in Appendix K.2 on Page 29 as follows:**
> > "As mentioned in Section 1, periodicity detection offers benefits for other time series-related tasks, such as classification, clustering, decomposition, anomaly detection, and forecasting. In this regard, we will now delve deeper into how accurately estimated periods can boost the performance of these tasks.
> > - Time series classification and clustering: Periodic patterns in time series data often contain informative and discriminative features that can distinguish different classes or clusters with the noise and irrelevant variations in time series filter out. For example, in heart rate monitoring, periodicity detection can identify recurring patterns, such as beat-to-beat intervals, enabling the differentiation between normal heart rhythms and irregularities related to cardiac conditions  (Thungtong et al., 2018). Similarly, in customer behavior analysis, periodicity detection helps identify regular patterns in purchasing behavior, facilitating the clustering of customers based on their shopping habits (Guidotti et al., 2018).
> > - Time series decomposition: Many typical time series decomposition methods, such as MSTL (Bandara et al, 2021) and robustSTL (Wen et al., 2019), rely on predefined periods for extracting seasonal components. By accurately detecting periodic patterns, we can extract underlying seasonal variations more effectively, leading to a cleaner and more accurate decomposition.
> > - Anomaly detection: Periodicity detection aids in accurately identifying and characterizing normal patterns or behaviors in time series data. Establishing a baseline for normal behavior based on the underlying periodic patterns enables the detection of anomalies or deviations from expected periodic behavior (Shehu et al, 2023). For instance, in network traffic analysis, periodicity detection helps identify regular communication and data transfer patterns, enabling the identification of anomalies that deviate from the expected periodic behavior (Akpinar et al., 2020).
> > - Time series forecasting: Periodicity detection allows for the identification and modeling of seasonal, trend, and noise patterns in time series data, as mentioned above. By accurately capturing the underlying periodic components, we can extract and model seasonal and trend variations more accurately, while reducing the influence of unpredictable noise. Periodicity patterns have been exploited explicitly or implicitly in various forecasting algorithms, including ARIMA (Box et al., 1968), prophet (Taylor et al., 2018), N-BEATS (Oreshkin et al., 2019), Pyraformer (Liu et al., 2021), TimesNet (Wu et al., 2023), etc."
>
>
> Moreover, **we have outlined future research directions in Appendix K.3 on Page 30, highlighting the exploration of additional downstream tasks as an interesting avenue to pursue**, as follows:
> > "In future research, an intriguing avenue to explore is the concept of pre-training in the context of time series analysis. Specifically, we can investigate the feasibility of pre-training an inference network using a large and diverse set of time series data. This pre-trained network can then be directly applied to perform periodicity detection without requiring additional training. Additionally, **it would be valuable to assess whether the features extracted by the pre-trained inference network are also effective for other time series-related tasks, such as classification and forecasting. To this end, the pre-trained inference network could be utilized for downstream tasks with some fine-tuning, further investigating its potential for enhancing performance across various time series analysis applications.**"

---

### Official Review · Reviewer_M5wN · 2023-11-01

**Soundness:** 3 good
**Presentation:** 2 fair
**Contribution:** 3 good
**Rating:** 6
**Confidence:** 3

**Summary:**

The authors propose a new approach for identifying periodic patterns in time-series data. They introduce AmortizedPeriod, which integrates Bayesian statistics and deep learning using an attention-based amortized variational inference model. AmortizedPeriod is capable of capturing various components of time series data, including multiple and irregular periods, trends, noise, outliers, and missing data. They also present a self/semi-supervised learning framework for the inference model to leverage knowledge from previously observed time series. Extensive experiments on four datasets demonstrate that AmortizedPeriod outperforms existing state-of-the-art methods with significant computational efficiency gains.

**Strengths:**

(1) The proposed model has good robustness in dealing with a variety of data anomaly scenarios, and addresses the limitations of current methods in terms of memorylessness. The authors present sufficient mathematical proofs in the method and appendix.
(2) The attention-based amortized variational inference model is innovative. The self/semi-supervised learning framework allows the inference model to leverage past knowledge, improving the inference efficiency on new data.
(3) The experiments implemented are relatively complete. From the experimental results, AmortizedPeriod has achieved a very significant advantage compared with state-of-the-art methods. The experimental section demonstrates the effectiveness and computational efficiency of AmortizedPeriod.

**Weaknesses:**

(1) The structure of thesis, however, is a little bit unreasonable. The description of the method takes up a lot of space, resulting in a somewhat inadequate analysis of the experiments that follow (although some are also presented in the appendix).
(2) The baselines in this work may not be enough. It is better to supplement some methods proposed in recent years to baselines. (e.g., Robust Dominant Periodicity Detection for Time Series with Missing Data", published at ICASSP 2023).
(3) To enhance the solidity of this work, it is suggested to add some public data sets (e.g., CRAN) to evaluate the performance on AmortizedPeriod and the baselines.

**Questions:**

(1) If CRAN is included, can it be described in two parts (single-periodicity detection and multi-periodicity detection) in the experimental analysis? I think that providing such an experimental analysis helps to further demonstrate the generalization of AmortizedPeriod.

---

> ### Author Response · Authors · 2023-11-23
> **Replay to Reviewer M5wN (Part 1)**
>
> Thanks very much for your detailed review and constructive feedback. We believe that addressing the reviewer’s comments has resulted in improving the clarity and presentation of the paper’s contributions and has brought the paper to a higher standard. We provide a detailed response to each of the reviewer's comments. The reviewer's comments are presented in italics, followed by our response. Quotations from the revised paper are included in markdown quotation mode. The corresponding modifications in the paper are highlighted in blue. Unless otherwise specified, all references to pages, equations, sections, and bibliographic citations relate to the revised paper.
>
> _Q1 - the structure of the thesis is a little bit unreasonable._
>
> Thank you for your feedback! We have taken your comment into consideration and made some adjustments to the structure of our paper. Firstly, to address the concern about the methodology section taking up a significant portion of the thesis, **we have moved the cumbersome definition of the Ramanujan subspaces to Appendix C. In Section 3.1, we now focus on discussing the rationale behind utilizing the Ramanujan subspaces for periodicity detection.**
>
> Furthermore, we have conducted additional experiments on the original four datasets. These experiments include evaluating the performance of benchmark methods using a broader range of metrics and highlighting the memory consumption of each method. Additionally, we have incorporated the experimental results for the CRAN dataset. However, due to space limitations, we were only able to present these additional experiments **in Appendix I and J (Pages 22-29). In the main body of the paper, we have appropriately referred to the appendix in Section 4 to provide readers with essential information about the experiments.**

---

> ### Author Response · Authors · 2023-11-23
> **Replay to Reviewer M5wN (Part 2)**
>
> _Q2 - add the CRAN dataset_
>
> Thanks for pointing this out! Originally, we did not include the CRAN data in our study due to the varying lengths of the time series in this dataset. AmortizedPeriod requires all time series to have equal lengths, making it incompatible with the CRAN dataset. Additionally, we considered single periods as a special case of multiple periods. Therefore, if AmortizedPeriod exhibits superior performance on datasets with multiple periods, it should also outperform other methods on datasets with single periods.
>
> However, after considering your feedback, **we recognize the importance of further demonstrating the generalization ability of AmortizedPeriod.** Therefore, we have made efforts to preprocess the CRAN dataset to ensure that all time series have a consistent length. The preprocessing steps are detailed in **Appendix J, located on Pages 27-28** as follows:
> > "It should be noted that the time series within this dataset have varying lengths. To enable the application of AmortizedPeriod to this dataset, a preprocessing step is undertaken involving linear interpolation and truncation. This ensures that all resulting time series have the same length of 1040. Specifically, if a time series $x^{\{i\}}$ has a length $L^{\{i\}}$ less than 1040, we perform linear interpolation by introducing $\lfloor 1040 / L^{\{i\}} \rfloor + 1$ new points between each pair of consecutive time points in the original series, where $\lfloor\cdot\rfloor$ represents the floor function. When $L^{\{i\}}$ is greater than or equal to 1040, we truncate the original series, retaining the first 1040 time points."
>
>
> **We then apply the four benchmark methods to the preprocessed CRAN dataset and present the results in Appendix J as follows:**
> > "The results of our experiments are presented in the following table, where it can be observed that the proposed AmortizedPeriod consistently outperforms other methods in terms of accuracy and efficiency across most evaluation criteria. It is worth noting that the performance of AmortizedPeriod on the testing data is comparatively lower than that on the training data. This discrepancy arises due to the limited number of time series in the training set (i.e., 64), which restricts the generalization capability of the learned inference network to unseen testing data. Nevertheless, we would like to emphasize that the periods for the training data are estimated in a self-supervised manner. Consequently, the superior $F_1$-score achieved by AmortizedPeriod indicates the versatility of the proposed generative model in describing the training data, while the inference model effectively maps the observed time series to the distribution of the periods.
>
> > On the other hand, the hyperparameters for AutoPeriod and RobustPeriod were set to be the same as those used in the four datasets discussed in Section 4. These parameters were manually tuned to yield good performance on those datasets. Although tuning these parameters specifically for the CRAN data might potentially enhance their performance, it would require additional effort. This underscores the importance of adaptive hyperparameter learning from the data, a feature effectively addressed by AmortizedPeriod.
>
> > Finally, for RPD, we employed cross-validation to select the single hyperparameter in all our experiments, which contributes to its relatively robust performance. However, since RPD does not account for the non-periodic components in the data, its performance remains inferior to that of AmortizedPeriod, particularly for the training data."
>
> >| Metrics | Datasets | AutoPeriod | RobustPeiod | RDP | AmortizedPeiod |
> >| --- | --- | --- | --- | --- | --- |
> >| Micro F1 | Train | 0.05 | 0.23 | 0.26 | **0.50** |
> >| Micro F1 | Test | 0.12 | 0.18 | 0.20 | **0.25** |
> >| Macro F1 | Train | 0.02 | 0.13 | 0.12 | **0.25** |
> >| Macro F1 | Test | 0.08 | **0.12** | 0.10 | **0.12** |
> >| Weighted F1 | Train | 0.05 | 0.42 | 0.49 | **0.51** |
> >| Weighted F1 | Test | 0.06 | 0.31 | **0.46** | 0.18 |
> >| Sampled F1 | Train | 0.02 | 0.25 | 0.34 | **0.50** |
> >| Sampled F1 | Test | 0.06 | 0.17 | **0.26** | 0.25 |
>
>
> _Q3 - baselines in recent years_
>
> Thank you for your valuable suggestion! Unfortunately, we were unable to find an open-source implementation of the algorithm mentioned in the paper published at ICASSP 2023. Instead, we have acknowledged this paper **in the related works section (Page 2)** and included it in **Table 4**, which summarizes all related works (**Page 15**).
>
> From our understanding, the novel approach presented in the ICASSP paper extends RobustPeriod to handle missing data. Considering that the proportion of missing data in our four datasets is relatively small, it is highly likely that the performance of RobustPeriod would be competitive with the novel approach. If so, the superiority of our proposed AmortizedPeriod over RobustPeriod may also extend to the novel approach.

---

### Official Review · Reviewer_Qy29 · 2023-11-07

**Soundness:** 3 good
**Presentation:** 2 fair
**Contribution:** 3 good
**Rating:** 5
**Confidence:** 2

**Summary:**

The paper studies the periodicity detection problem with the proposed AmortizedPeriod method by considering Bayesian modeling and deep learning. The motivation is good, the literature review is comprehensive, and the proposed method has competitive experimental performance. On the other side, some concerns are raised about the paper writing and the technical design and illustration. For more details, please refer to the following sections.

**Strengths:**

- The literature review is quite comprehensive, including the periodic time series, Ramanujan subspaces, and different kinds of priors.

- The Bayesian modeling designed for periodicity identification seems novel and comprehensive.

- The experiments are extensive, and the provided results are very competitive.

**Weaknesses:**

- The first and biggest concern raised by the reviewer is how the proposed two challenges get addressed by the proposed model. For the first challenge "robustness", how do the designed Bayesian modeling and encoder-decoder model address it? More importantly, how does the proposed AmortizedPeriod solve the "memorylessness" challenge, i.e., the second challenge, especially when the experimental running time of the proposed AmortizedPeriod is superior?


- Section 3.1 is somehow deep for some audiences and does not motivate Section 3.2 very well, the plain definition occupies much space but neither motivates the formulation proposal in Section 3.2 nor adds values for the statement of why it is sensitive to trends, noises, and outliers.


- The setting of semi-supervised learning in Section 3.5 is not very clear, e.g., what is the role of sub-period and how it works in the learning and classification.


- The symbols, notions, and indexing are somehow messy.

**Questions:**

Please refer to the first and the third points in the above section.

---

> ### Author Response · Authors · 2023-11-23
> **Reply to Reviewer Qy29 (Part 1)**
>
> Many thanks for your valuable feedback! We believe that addressing the reviewer’s comments has resulted in improving the clarity and presentation of the paper’s contributions and has brought the paper to a higher standard. We provide a detailed response to each of the reviewer's comments. The reviewer's comments are presented in italics, followed by our response. Quotations from the revised paper are included in markdown quotation mode. The corresponding modifications in the paper are highlighted in blue. Unless otherwise specified, all references to pages, equations, sections, and bibliographic citations relate to the revised paper.
>
>
> _Q1 - how AmortizedPeriod address the problem of "limited robustness" and "memorylessness"_
>
> AmortizedPeriod effectively addresses the problem of limited robustness, by considering all possible components in a time series (i.e., multiple and irregular periods, trends, noise, outliers, and incomplete observations) using a Bayesian model in a joint and probabilistic manner. This joint and probabilistic framework allows us to infer periodicity while simultaneously accounting for other components and their uncertainties. Moreover, Additionally, by involving the hyperparameters characterizing these components in the Bayesian model, we can adaptively infer their posterior distribution from the data instead of relying on manual specification, thus further enhancing the robustness of our method.
>
> On the other hand, AmortizedPeriod addresses the problem of memorylessness by learning a inference model that directly maps the observed time series to the distribution of the periodic components in a self or semi-supervised manner. Unlike existing methods such as RobustPeriod or RPD that require iterative optimization to estimate periods for new time series, AmortizedPeriod can input the new time series directly into the trained inference model to obtain the distribution of periods. This streamlined approach not only reduces computational time but also allows for the utilization of labeling information during training, as the inference model has the capability to "memorize" the labels.
>
> **Originally, we have explained how the proposed AmortizedPeriod address these two issues in the last paragraph of Section 1 on Page 2.**
>
> Moreover, after carefully considering the feedback from reviewer Qy29, we believe it is beneficial to further discuss how AmortizedPeriod tackles these problems after the literature review section (i.e., Section 2). This placement allows readers to develop a clearer understanding of how AmortizedPeriod overcomes the limitations encountered in previous works. Therefore, **we have included an additional paragraph at the beginning of Section 3 on Page 3**, as follows:
>
> > "The limited robustness observed in previous studies can be attributed to the neglect of non-periodic components, such as trends, noise, and outliers. While certain methods, like RobustPeriod, acknowledge these components, they eliminate them individually in a deterministic and sequential way, leading to error accumulation. Moreover, the performance of each stage is sensitive to the choice of the corresponding hyperparameters. To overcome these difficulties, we propose a compact Bayesian generative model that offers flexibility in modeling the interaction between all components as well as their corresponding hyperparameters in a joint and probabilistic manner. On the other hand, existing approaches derive period estimates for each time series from scratch by solving optimization problems iteratively. Moreover, even if we correct the estimated periods for a time series, applying existing methods would still yield the same erroneous results. To provide a remedy, we develop an attention-based inference network that directly maps observed time series to the posterior distribution of different components and the hyperparameters, including the distribution of the periods. As a result, the inference model can provide period distributions for new time series without iterative optimization. Moreover, the inference model can be trained using labeling information, as it can memorize the labels. In the sequel, before delving into the generative and inference model, we first introduce the Ramanujan subspaces, which form the foundation of AmortizedPeriod."

---

> ### Author Response · Authors · 2023-11-23
> **Reply to Reviewer Qy29 (Part 2)**
>
> _Q2 - Section 3.1 fails to motivate the following sections_
>
> Thanks a lot for pointing this out! We acknowledge that Section 3.1 could benefit from better organization. As a result, we have made several revisions. **Firstly, we have moved the definition of the Ramanujan subspaces to Appendix C. In Section 3.1, we now focus on explaining the rationale behind using the Ramanujan subspaces for periodicity detection**, as follows:
>
> > "The definition of Ramanujan subspaces is cumbersome to describe and not illuminating, so we present instead the definition of $q$-periodic time series as a starting point. From there, we proceed to discuss the properties of Ramanujan subspaces relevant to periodicity detection and define exactly $q$-periodic time series. We refer the readers to Appendix C for the formal definition of Ramanujan subspaces.
>
> > Defintion 1. _($q$-periodic) A time series $s(t)$ with time stamp $t$ of length $L$ ($1\leq t \leq L$) is $q$-periodic if $q$ is the smallest integer such that $s(t + q) = s(t)$ for all $t$, and $q$ is called the period of $s(t)$_.
>
> > This definition eliminates the ambiguity that a time series of period $q$ is also of a period that is a multiple of $q$, which plagues the time-domain methods for periodicity detection. However, another problem comes along with this definition, that is, a time series with an exact period of 6, for instance, may be further decomposed into two periodic components with periods 2 and 3 respectively, making it necessary to detect these fundamental exact periods in practical scenarios. To attack this problem, Vaidyanathan (2014a;b) proposed the Ramanujan subspaces, which construct a distinct linear subspace $\mathcal S_q$ for each period $q$. The time series spanned by a subspace $\mathcal S_q$ can only possess a period of $q$ and cannot be decomposed into smaller periods. Moreover, two arbitrary subspaces $\mathcal S_{q_1}$ and $\mathcal S_{q_2}$ are orthogonal to each other if $q_1 \neq q_2$. As a result, we can define the exact period as:
>
> > Definition 2. _(exactly $q$-periodic (Muresan & Parks, 2003)) A time series $s(t)$ is of exact period $q$ if  $s(t)$ is in the Ramanujan subspace $\mathcal S_q$ and the projection of $s(t)$ onto_ $\mathcal S_\bar{q}$ _is zero for all $\bar q < q$._
>
> > Let $R_q$ be the linear basis for $\mathcal S_q$ and $D = [R_1, \cdots, R_{P_\text{max}}]$ the resulting Ramanujan dictionary with a maximum period of $P_\text{max}$. We can ascertain the periods of a time series $s(t)$ by decomposing it as $s = D \alpha$, where $\alpha$ is the coefficient vector. Ideally, $\alpha$ should have non-zero entries only at locations where the periodic components of $s$ reside. Estimating $\alpha$ given $s$ and $D$ can be formulated as a sparse vector recovery problem, such as basis pursuit or lasso (Tenneti & Vaidyanathan, 2016)."
>
>
> **To better motivate the subsequent sections, we have also revised the first paragraph of Section 3.2 as follows:**
>
> > "Unfortunately, the above formulation is susceptible to the presence of trends, noise, outliers, and incomplete observations that often coexist with periodic components in time series, as well as the hyperparameter promoting sparsity in basis pursuit or lasso. Conversely, Bayesian models or networks offer a compact, flexible, and interpretable integration of various components and their associated hyperparameters in time series, while also accounting for the accompanying uncertainty. However, their application in the context of periodicity identification remains unexplored. To bridge the gap, we propose a reformulation of the periodicity detection problem from the Bayesian perspective."
>
> **Finally, since we have ample space available in the appendix, we provide a more detailed explanation of the definitions of Ramanujan subspaces and their properties that are relevant to periodicity detection.** For more information, please refer to Appendix C.

---

> ### Author Response · Authors · 2023-11-23
> **Reply to Reviewer Qy29 (Part 3)**
>
> _Q3 - the setting of semi-supervised learning in Section 3.5 is not very clear_
>
> As clarified in **the second to last paragraph of Appendix C on Page 16**:
> > "Consider the sum $x(t) = \sum_i x_{q_i}(t)$, where $x_{q_i}(t)$ is an arbitrary $q_i$-period time series (cf. Definition 1), the period of $x(t)$ is equal to the least common multiple (LCM) of $q_i$, i.e., $\text{LCM}(q_i)$, or a divisor of $\text{LCM}(q_i)$, as proven in (Vaidyanathan, 2014a; Tenneti & Vaidyanathan, 2015). For instance, given two 2-periodic time series $\{1,-1,1,-1,\cdots\}$ and $\{-1,1,-1,1,\cdots\}$, the period of their sum is 1, which is a divisor of $\text{LCM}(2,2) = 2$. However, if the exact period of $x_{q_i}(t)$ is $q_i$, that is, $x_{q_i}(t) \in \mathcal S_{q_i}$ , the period of $x(t)$ can only be $\text{LCM}(q_i)$ and not smaller. This is another attractive property of the Ramanujan subspace. Conversely, any arbitrary $q$-periodic time series, as defined in Definition 1, can be spanned by the set of Ramanujan subspaces ${\mathcal S_{q_i}}$, where $q_i$ represents all divisors of $q$, including both 1 and $q$. For example, a 6-periodic time series can be obtained by adding periodic components from $\mathcal S_2$ and $\mathcal S_3$, $\mathcal S_1$ and $\mathcal S_6$, or $\mathcal S_1$, $\mathcal S_2$, $\mathcal S_3$, and $\mathcal S_6$, as long as the LCM of the exact periods corresponding to these subspaces equal 6. In other words, we can summarize this concept with the following proposition:
>
> > Proposition 2. _**According to the definition of periods in Definition 1 and exact periods in Definition 2, the period of a time series is equal to the least common multiple (LCM) of its exact periods.**_"
>
>
> Given that the period of a time series is $q$, it may be spanned by the Ramanujan subspaces $\mathcal S_q$ and the subspaces $\mathcal{S}_{d_q}$ corresponding to the divisors of $q$. Therefore, the coefficients with respect to these subspaces may be non-zero, while the coefficients with respect to the remaining periods must be zero. We incorporate this information into the original Evidence Lower Bound (ELBO) in a semi-supervised manner, as explained in Eq. (12).
>
> Due to the page limit, we explicitly mention in Section 3.4 on Page 7 that
> > "**For a time series with period of $q$, it is likely that all bases within the Ramanujan subspaces associated with $q$ and its divisors $d_q$ will be selected, as the period of a time series equals the LCM of the exact periods of all its periodic components (cf. Proposition 2)**"
> and then direct the readers to Proposition 2 and Appendix C.
>
> _Q4 - notations are messy_
>
> Thanks for pointing this out! We have added a table in the appendix (see **Table 3 in Appendix A on Page 14**) that provides a comprehensive summary of all the notations used in this paper. We hope that this table will be helpful to readers in clarifying any confusion regarding the definition of the notations.

---

### Meta-Review · Area_Chair_LbSc · 2023-12-14

**Metareview:**

The authors propose a new approach for identifying periodic patterns in time-series data. AmortizedPeriod uses an attention-based amortized variational inference model to address challenges of limited robustness and lacking memory in existing methods. Emprical validation demonstrates that the proposed method outperforms SOTA with computational efficiency gains.

Strength: the reviewers find the method innovative, improved robustness to handle multiple and irregular periods, trends, noise, outliers, and missing data is a great property. The experiments are extensive and showed competitive performance.

Weakness: the writing can be improved. Reviewers also asked for more comparison to recent baselines as well.

**Justification For Why Not Higher Score:**

The writing can be improved. Reviewers also asked for more comparison to recent baselines.

**Justification For Why Not Lower Score:**

The reviewers find the method innovative, improved robustness to handle multiple and irregular periods, trends, noise, outliers, and missing data is a great property. The experiments are extensive and showed competitive performance. The rebuttal was able to address majority of reviewer concerns.

---

### Decision · Program_Chairs · 2024-01-16

Accept (poster)